# Response repetition biases in human perceptual decisions are explained by activity decay in competitive attractor models

James J Bonaiuto*, Archy de Berker, Sven Bestmann

Sobell Department of Motor Neuroscience and Movement Disorders, UCL Institute of Neurology, University College London, London, United Kingdom

**Abstract** Animals and humans have a tendency to repeat recent choices, a phenomenon known as choice hysteresis. The mechanism for this choice bias remains unclear. Using an established, biophysically informed model of a competitive attractor network for decision making, we found that decaying tail activity from the previous trial caused choice hysteresis, especially during difficult trials, and accurately predicted human perceptual choices. In the model, choice variability could be directionally altered through amplification or dampening of post-trial activity decay through simulated depolarizing or hyperpolarizing network stimulation. An analogous intervention using transcranial direct current stimulation (tDCS) over left dorsolateral prefrontal cortex (dlPFC) yielded a close match between model predictions and experimental results: net soma depolarizing currents increased choice hysteresis, while hyperpolarizing currents suppressed it. Residual activity in competitive attractor networks within dlPFC may thus give rise to biases in perceptual choices, which can be directionally controlled through non-invasive brain stimulation.

*For correspondence:
j.bonaiuto@ucl.ac.uk

**Competing interests:** The authors declare that no competing interests exist.

## Introduction

Perceptual and value-based decisions in humans and animals are often characterized by choice biases (*Hunt, 2014*; *Nicolle et al., 2011*; *Fleming et al., 2010*; *Padoa-Schioppa, 2013*; *Noorbaloochi et al., 2015*; *De Martino et al., 2006*; *Chen et al., 2006*; *Rorie and Newsome, 2005*; *Tom et al., 2007*). For example, human and nonhuman primate value choices are subject to various biases such as framing effects (*De Martino et al., 2006*), choice repetition biases (*Padoa-Schioppa, 2013*; *Samuelson and Zeckhauser, 1988*), sunk cost effects (*Bogdanov et al., 2015*), and previous payoff biases (*Noorbaloochi et al., 2015*; *Rorie et al., 2010*). These biases, which become more pronounced with difficult decisions (*Fleming et al., 2010*; *Padoa-Schioppa, 2013*), are also observed in human perceptual decision making (*Nicolle et al., 2011*; *Fleming et al., 2010*; *Noorbaloochi et al., 2015*; *Mulder et al., 2012*; *St John-Saaltink et al., 2016*; *Akaishi et al., 2014*), with correlational evidence for a link between neural and choice variability (*St John-Saaltink et al., 2016*; *Hesselmann et al., 2008*; *Wyart and Tallon-Baudry, 2009*). Choice repetition biases are especially intriguing because they provide a window on decision making outside of the laboratory. In real life, decisions do not occur in discrete and isolated trials with long inter-trial intervals, but rather take place within the context of, and are therefore potentially biased by, previous decisions made in the immediate past.

This notion has indeed been elegantly recognized in recent economic decision making work in non-human primates, suggesting that decaying trace activity from the previous choice in competitive neural circuits increases the likelihood of repeating that choice when there is a small subjective

**eLife digest** When making decisions, people and other animals tend to repeat previous choices even if this is no longer the best course of action. This tendency is especially common when the choice is difficult to make. For example, when people are asked to decide whether groups of dots on a television screen are moving mostly to the left or the right, they often repeat their previous choice when the direction of motion is not clear.

Recordings of brain activity in animals suggest that once a choice is made, there is brain activity left over that influences the level of activity at the beginning of the following choice. If this leftover activity is stronger in the brain cells that represent the first choice, it might give this option a head start when another decision is made; this would provide one explanation as to why that same choice is repeated. However, this explanation had not been tested directly.

Bonaiuto et al. reasoned that if leftover activity is indeed the cause of choice repetition, directly manipulating this activity in the human brain should alter this tendency in a predictable way. First, computer-based simulations of circuits of brain cells were used to predict what the consequences of such manipulation would be. The model predicted that brain activity left over after a choice is made would indeed cause the choice to be repeated. Moreover, stimulating this virtual circuit did increase or decrease the tendency to repeat choices depending on the type of stimulation used.

Bonaiuto et al. went on to confirm that human volunteers who had been asked to complete the "moving dots" task did tend to repeat their choices. Next, the volunteers had a region of their brain, which is known to be important for making choices, stimulated using electrodes placed on their scalp (a non-invasive method of brain stimulation). Exactly as the computer simulations predicted, one form of stimulation made the individual more likely to repeat their previous choice, while another form of stimulation had the opposite effect.

These findings show that stimulating the brain via a non-invasive technique can shape the choices that people make in ways that can be predicted by a biologically realistic computer simulation of networks in the brain. The findings also support the idea that leftover activity following a choice might be the biological reason why people tend to go against evidence and repeat previous choices. This new knowledge could be exploited in future studies that try to understand and influence decision making in humans.

difference between the current decision options (*Padoa-Schioppa, 2013*). These results have been explained by sustained recurrent activity in competitive attractor networks, which gradually returns to baseline levels following a decision, but can bias activity in the following trial in tasks with short inter-trial intervals (*Rustichini and Padoa-Schioppa, 2015*). Indeed, a series of studies have linked perceptual and value-based decision-making with activity in such competitive attractor networks (*Rustichini and Padoa-Schioppa, 2015*; *Hunt et al., 2012*; *Bonaiuto and Arbib, 2014*; *Hämmerer et al., 2016*; *Wang, 2008*; *Wong et al., 2007*; *Wang, 2012, 2002*; *Martí et al., 2008*; *Mazurek et al., 2003*; *Moreno-Bote et al., 2007*; *Deco and Rolls, 2005*; *Deco et al., 2009*; *Usher and McClelland, 2001*; *Bogacz et al., 2007*; *Furman and Wang, 2008*; *Deco et al., 2013*; *Braun and Mattia, 2010*; *Jocham et al., 2012*). Here, we show that carry-over activity in these networks produces a conspicuous bias to repeat difficult choices which is mirrored in the behavior of human participants. We further show that a characterization of this phenomenon in silico allows us to make directional predictions of the effects of transcranial stimulation upon choice bias which are further borne out by behavioural experiments.

Specifically, we used a combination of human experimentation and computational modeling to investigate the mechanisms underlying choice hysteresis during perceptual decision making. We used an established and biophysically plausible model of a decision making network that employs competition between neural populations to choose between two alternate response options. Rather than simulating discrete trials and reinitializing the network state at the start of each trial, we sought to emulate the serial dependency between real world choices. We therefore ran the network in continuous blocks of trials, with the final state at the end of each trial serving as the initial state of the next trial (*Rustichini and Padoa-Schioppa, 2015*). We confirmed that this produced choice

hysteresis in the model behavior, through decaying trace activity from the previous trial biasing selection in the current trial, but only for short inter-stimulus intervals. We then conducted an analogous experiment with human participants and found a similar tendency to repeat previous choices.

The model contains variables and parameters with well-defined anatomical and physiological substrates (*Rustichini and Padoa-Schioppa, 2015*; *Bonaiuto and Arbib, 2014*; *Wang, 2008*, *2012*, *2002*), allowing for explicit simulation and linkage with the known neurophysiological effects of stimulation. We found that perturbation of the model's trace activity through simulated changes in the network's membrane potential led to predictable alterations in choice bias. In human participants, we therefore applied transcranial direct current stimulation (tDCS) to left dorsolateral prefrontal cortex (dlPFC), a region implicated in perceptual decision making (*Heekeren et al., 2004*; *Kim and Shadlen, 1999*; *Heekeren et al., 2006*; *Philiastides et al., 2011*; *Rahnev et al., 2016*; *Georgiev et al., 2016*). TDCS is thought to alter neuronal excitability and spontaneous firing rates in brain networks by polarizing membrane potentials in a network (*Rahman et al., 2013*; *Nitsche and Paulus, 2011*; *Bikson et al., 2004*), thus providing an analogous network perturbation to our simulations. Because tDCS leads to subthreshold polarization changes, we were able to subtly alter the spontaneous fluctuations in neural activity within the targeted brain region, noninvasively in our human participants (*Nitsche and Paulus, 2011*, *2000*; *Kuo and Nitsche, 2012*).

We found that the predictions generated by the model were closely mirrored by the modulation of choice hysteresis in human participants through application of tDCS over dlPFC. We were thus able to directionally control choice biases in perceptual decision making through causal manipulation of the neural dynamics in dlPFC. The comparison with the model suggests that this control of choice hysteresis arises from an amplification or suppression of sustained recurrent activity, which biases the following decision.

## Results

### Competitive attractor model architecture

We used an established spiking neural model of decision making implementing an attractor network (*Bonaiuto and Arbib, 2014*; *Wang, 2008*; *Wong et al., 2007*; *Wang, 2012*, *2002*; *Deco et al., 2009*; *Bonaiuto and Bestmann, 2015*; *Rolls et al., 2010*; *Wong and Wang, 2006*; *Lo and Wang, 2006*; *Machens et al., 2005*). This model was initially developed to explain the neural dynamics of perceptual decision making and working memory (*Wang, 2002*) and has been used to investigate the behavioral and neural correlates of a wide variety of perceptual and value-based decision making tasks at various levels of explanation (*Rustichini and Padoa-Schioppa, 2015*; *Hunt et al., 2012*; *Bonaiuto and Arbib, 2014*; *Hämmerer et al., 2016*; *Wang, 2012*, *2002*; *Furman and Wang, 2008*; *Jocham et al., 2012*; *Bonaiuto and Bestmann, 2015*; *Rolls et al., 2010*; *Wong and Wang, 2006*). The model is well suited for computational neurostimulation studies because it is complex enough to simulate network dynamics at the neural level, yet is simple enough to generate population-level (neural and hemodynamic) signals, and the resulting behavior allows for comparison with human data (*Hunt et al., 2012*; *Bonaiuto and Arbib, 2014*; *Rolls et al., 2010*). The model also incorporates neurons at a level of detail that allows simulation of tDCS by the addition of extra transmembrane currents with parameter values comparable to previous modeling work (*Hämmerer et al., 2016*; *Bonaiuto and Bestmann, 2015*; *Molaee-Ardekani et al., 2013*), and current understanding of the mechanism of action of tDCS (*Rahman et al., 2013*; *Nitsche and Paulus, 2011*; *Bikson et al., 2004*; *Funke, 2013*; *Radman et al., 2009*; *Bindman et al., 1964*).

The model consists of two populations of pyramidal cells representing the available response options, which are 'left' and 'right' in this task (*Figure 1A*). Each population receives task-related inputs signaling the perceived evidence for each response option. The difference between the inputs varies inversely with the difficulty of the task (*Figure 1A*, inset), and the rate of each input is sampled according to the refresh rate of the monitor used in our experiment (60 Hz, *Figure 1B*, left column). The pyramidal populations are reciprocally connected and mutually inhibit each other indirectly via projections to and from a common pool of inhibitory interneurons. This pattern of connectivity gives rise to winner-take-all behavior in which the firing rate of one pyramidal population (typically the one receiving the strongest inputs) increases and that of the other is suppressed, indicating the decision.

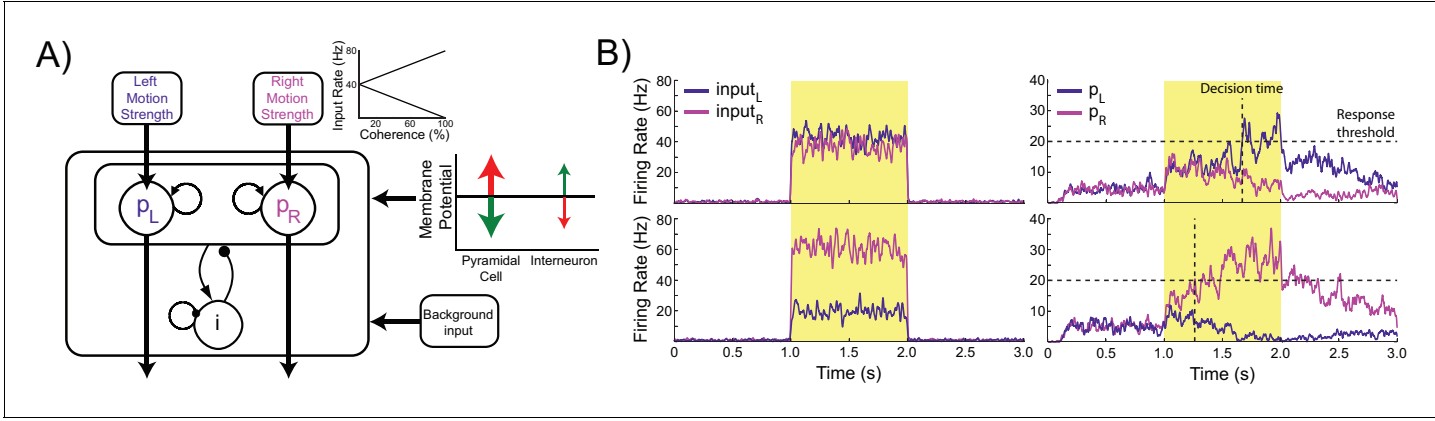

**Figure 1.** Model architecture. (**A**) The model contains two populations of pyramidal cells which inhibit each other through a common pool of inhibitory interneurons. The pyramidal populations receive task-related inputs signaling the momentary evidence for each response option. The mean input firing rate to each pyramidal population varies as a function of the stimulus coherence (inset). Difficult trials have low coherence, easy trials high coherence. tDCS is simulated by modulating the membrane potential of the pyramidal and interneuron populations. (**B**) Firing rates of the task-related inputs (left column) and two pyramidal populations (right column) during representative trials with low (top row), and high (bottom row) coherence. The horizontal dotted lines denote the response threshold (20 Hz in this example) and the vertical dotted lines show the decision time - when one of the pyramidal population's firing rate crosses the response threshold.

In difficult trials each input fires at approximately the same rate, while in easy trials one input fires at a high rate while the other fires at a very low rate (*Figure 1B*, right column).

## Post-trial residual firing in an attractor network produces choice bias on the current trial

We simulated behavior in a perceptual decision making task by scaling the magnitude of the task-related inputs to emulate input from a virtual Random Dot Kinetogram (RDK) with varying levels of coherent motion. The behavior was produced by virtual subjects, which were created by instantiating the model with parameters sampled from distributions designed to capture between-participant variability in human populations (see Materials and methods). In order to analyze the behavioral output of the network, we consider a response option to be *chosen* when the corresponding pyramidal population exceeds a set *response threshold*. We measured the accuracy of the model's performance as the percentage of trials in which the chosen option corresponded to the stronger task-related input. For comparison between virtual subjects and human participants, we defined the *accuracy threshold* as the coherence level required to attain 80% accuracy. The time step at which the response threshold is exceeded is taken as the *decision time* for that trial (*Figure 1B*). Because we do not simulate perceptual and motor processes involved in encoding visual stimuli and producing a movement to indicate the decision, this is distinct from the *response time* measured in human participants.

As expected, the model generates increasingly accurate responses at higher coherence levels (*Figure 2A*). This is because the 'correct' pyramidal population is receiving much stronger input than the other, allowing it to more easily win the competition by exerting strong inhibitory influence onto the other pyramidal population pool. In line with previous work, the model predicts a decrease in decision time with increasing coherence (*Wang, 2002*) (*Figure 2B*). In terms of model dynamics, when motion coherence is low the sensory evidence for the left and right choices is approximately equal, and therefore the inputs that drive both pyramidal populations are more balanced. As a consequence it takes longer for one population to 'win' over the other and for the network to reach a stable state (*Figure 1B*).

Turning to our main question about choice biases, we simulated performance of the task by running the model in a continuous session (*Figure 2D*). Thus, rather than resetting the model state at the start of each trial, as in previous work (*Bonaiuto and Arbib, 2014*; *Hämmerer et al., 2016*; *Wang, 2002*; *Bonaiuto and Bestmann, 2015*), we used the network state at the end of the previous

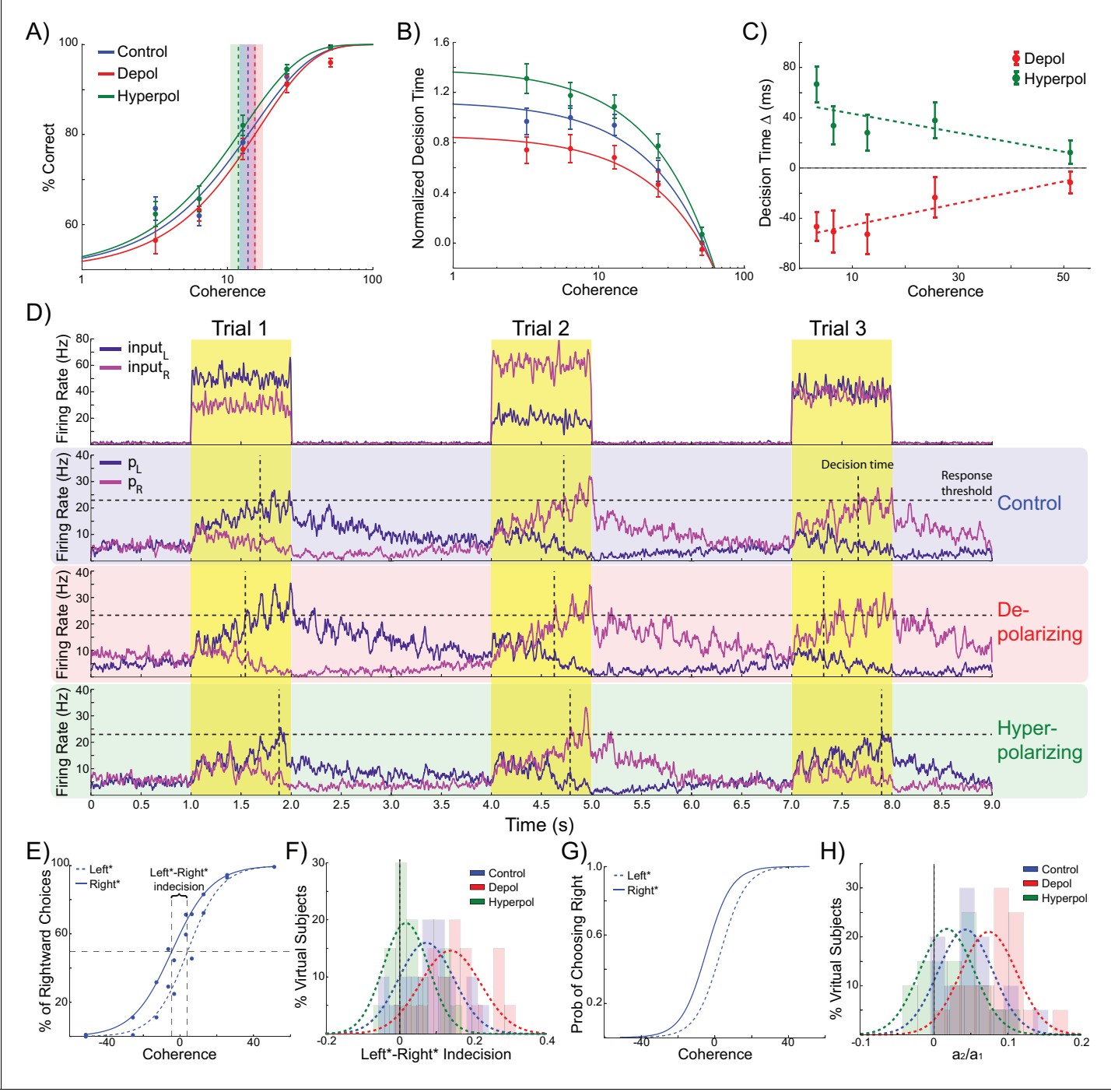

**Figure 2.** Effects of simulated network stimulation on model behaviour. (A) There was no average change in the decision threshold with either depolarizing or hyperpolarizing stimulation, where the decision threshold reflects the coherence required to reach 80% accuracy. (B) Decision time decreases with increasing coherence, with depolarizing stimulation speeding decision time and hyperpolarizing stimulation slowing decisions. (C) Depolarizing stimulation decreases and hyperpolarizing stimulation increases decision time, but this effect is reduced with increasing coherence. (D) Neural dynamics of the model. The model was run continuously, with the decaying activity of each trial influencing the initial activity at the beginning of the following trial. Depolarizing stimulation delayed the return of this decaying activity to baseline levels, while hyperpolarizing stimulation dampened the overall dynamics of the model and therefore suppressed residual activity. (E) When sorted by the choice made on the previous trial (Left* or Right*), the indecision point (or level of coherence resulting in chance selection of the same choice), shifts. This reflects a bias towards repeating that decision. (F) The positive shift in indecision point is further increased by depolarizing stimulation and decreased by hyperpolarizing stimulation. (G) A logistic regression model was fit to choice behavior with coefficients for coherence and the choice on the previous trial. (H) This analysis confirms a positive

*Figure 2 continued on next page*

*Figure 2 continued*

value for the influence of the previous choice on the current choice (a₁), scaled by the influence of coherence (a₂). Depolarizing stimulation increases this ratio, and hyperpolarizing stimulation reduces it. See *Figure 2—source data 1* for raw data.
The following source data is available for figure 2:

**Source data 1.** Competitive attractor model accuracy, decision time, and choice hysteresis with simulated network stimulation.

trial as the starting state of the next trial (*Rustichini and Padoa-Schioppa, 2015*). The network displays sustained recurrent dynamics, due in large part to the slow time constants of the NMDA receptors modeled in the pyramidal cell populations. As a consequence, residual activity from the previous trial may still influence the dynamics of the network when the task-related inputs of the subsequent trial arrive (*Figure 2D*). We next asked if the model behavior exhibited any choice hysteresis, and whether this systematically related to any neural hysteresis effects.

We analyzed possible choice hysteresis effects in the model behavior by separating trials into two groups based on the decision made in the previous trial (Left*: trials where left was chosen in the previous, and Right*: trials following rightward choices). For each group we then fit the percentage of rightward choices to a sigmoid function of the coherence to the left or right (*Padoa-Schioppa, 2013*; *Rustichini and Padoa-Schioppa, 2015*). We found that this choice function was shifted according to the previously selected direction, reflecting a tendency to repeat the previous choice. This effect was particularly pronounced during difficult trials (*Figure 2E*). We defined the 'indecision point' as the level of coherence where rightward choices were made 50% of the time, and compared this value between Left* and Right* trials for each virtual subject across stimulation conditions. The model predicts a significant shift in indecision point depending upon the choice made in the previous trial (W(19) = 21, p=0.002; *Figure 2F*). This result was confirmed with a logistic regression analysis which more precisely accounted for the relative influences of current trial coherence and previous choice on decisions (*Padoa-Schioppa, 2013*; *Rustichini and Padoa-Schioppa, 2015*) (*Figure 2G*), and again found a significant influence of the previous choice on the decision (W(19) = 10, p<0.001; *Figure 2H*).

## Perturbation of an attractor network modulates choice hysteresis

The model suggests that biases in decaying tail activity from the previous trial can cause choice hysteresis. One would then expect that perturbation of the neural dynamics of the model alters hysteresis biases in a systematic way. We therefore asked how stimulation of our model altered its dynamics, and how these influence the model's behavior. We injected an additional trans-membrane current into pyramidal cells and inhibitory interneurons, with the polarity and magnitude based on simulations that reproduce tDCS-induced changes in sensory evoked potentials (*Molaee-Ardekani et al., 2013*) and behavior (*Bonaiuto and Bestmann, 2015*) in vivo, and taking into account the cellular effects of tDCS (*Hämmerer et al., 2016*; *Rahman et al., 2013*; *Nitsche and Paulus, 2011*; *Bikson et al., 2004*; *Bonaiuto and Bestmann, 2015*; *Funke, 2013*; *Radman et al., 2009*; *Bindman et al., 1964*). One advantage of combining experimental human studies with computational models is that it allows for interrogation of the putative neural dynamics of the model under different experimental manipulations (*Hämmerer et al., 2016*; *Bonaiuto and Bestmann, 2015*; *Fröhlich, 2015*; *Bikson et al., 2015*; *Bestmann, 2015*; *de Berker et al., 2013*).

Relative to no stimulation, there was no effect of depolarizing or hyperpolarizing stimulation on the model's accuracy threshold (depolarizing: W(19) = 65, p=0.135; hyperpolarizing: W(19) = 74, p=0.247; *Figure 2A*). This is consistent with previous work showing that for low levels of stimulation intensity (such as that used in these simulations), the resulting shifts in membrane potential are insufficient to completely reverse the model dynamics such that it significantly alters choice accuracy (*Bonaiuto and Bestmann, 2015*). However, we found that depolarizing stimulation decreased decision time, whilst hyperpolarization increased it (*Figure 2B*). We then analyzed the difference in decision time between no stimulation and stimulation conditions at each motion coherence level. In both stimulation conditions, this difference is strongest for difficult, low coherence trials, as indicated by the significant slopes in the linear fits between coherence and decision time difference (depolarizing: $B_1$ = 89.251, p=0.017; hyperpolarizing: $B_1$ = −77.327, p=0.034; *Figure 2C*). This is because during

difficult trials (low coherence), shifts in membrane potential induced by depolarizing stimulation cause the winning population to reach the response threshold earlier, compared to no stimulation, while hyperpolarizing stimulation delays this event. However, during high coherence trials, the strong difference in task-related input strengths overwhelms the subtle effects of membrane potential changes. These simulations therefore predict that response time should be unaffected by subtle changes in network dynamics caused by stimulation on 'no brainer' trials in which strong inputs provide unequivocal evidence for one response over the other. This echoes findings from human experiments that tDCS may interact with task difficulty and/or individual differences in performance (*Benwell et al., 2015*; *Jones and Berryhill, 2012*). The model thus predicts that network stimulation will affect response time, especially in difficult trials, but leave accuracy largely unaffected. It is predicted that depolarizing and hyperpolarizing stimulation will lead to faster and slower responses, respectively. We obtained qualitatively similar results in simulations controlling for the input parameters and effects of stimulation on interneurons, but not those that violate the known neural effects of stimulation (*Tables 1* and *2*, see Materials and methods).

In addition to decision time, we found significant effects of model stimulation on choice hysteresis. Depolarizing stimulation increased the indecision point shift, relative to no stimulation, (W(19) = 44, p=0.023), whereas hyperpolarizing stimulation decreased it (W(19) = 41, p=0.017). This result was echoed in a logistic regression analysis, which showed that depolarizing stimulation increased the relative influence of the previous choice to coherence (W(19) = 32, p=0.006), while hyperpolarizing stimulation reduced this ratio (W(19) = 42, p=0.019). In other words, the model demonstrated that choice hysteresis is caused by residual activity from the previous trial. Moreover, depolarizing stimulation increases this residual activity, while hyperpolarizing stimulation suppresses it. These results were replicated in alternative simulations using similar assumptions about the effects of stimulation, but not in those where the initial state of the network is reset at the start of each trial, or where the effects of stimulation were qualitatively different (*Table 3*, see Materials and methods).

As can be seen in *Figure 3*, each pyramidal population fires at approximately 3–15 Hz prior to the onset of the task-related inputs. We sorted the population firing rates of each trial based on which pyramidal population was eventually chosen, and then split trials into those in which the previous choice was repeated and those where a different choice was made. We found that in trials in which the previous choice was repeated, the mean firing rate of the chosen population was slightly higher than that of the unchosen population prior to onset of task-related input (*Figure 3A,C*). This effect can be attributed to decaying tail activity from the previous trial, which we refer to as *hysteresis bias*. This bias was amplified by depolarizing (W(19) = 4, p<0.001) and attenuated by hyperpolarizing stimulation (W(19) = 6, p<0.001; *Figure 3C*). The network was only able to overcome the bias and make a different choice from the one it made in the previous trial when the bias was very small and the model activity was dominated by the task-related inputs (*Figure 3B,D*).

If decaying tail activity in the chosen pyramidal population from the previous trial causes behavioral choice hysteresis effects, these effects should diminish with longer inter-stimulus intervals (ISIs). Given a long enough ISI, residual pyramidal activity is more likely to fully decay back to baseline

**Table 1.** Accuracy threshold statistics.

| | Depolarizing | | Hyperpolarizing | |
|---|---|---|---|---|
| | W(19) | p | W(19) | P |
| Total task-related input firing rates = 60 Hz | 62 | 0.108 | 67 | 0.156 |
| Refresh rate = 30 Hz | 69 | 0.179 | 57 | 0.073 |
| Refresh rate = 120 Hz | 34 | 0.008 | 78 | 0.314 |
| Inhibitory interneuron stimulation | 76 | 0.279 | 92 | 0.627 |
| Pyramidal cell stimulation only | 91 | 0.601 | 89 | 0.55 |
| Uniform stimulation | 84 | 0.433 | 43 | 0.021 |
| Reinitialization | 81 | 0.37 | 91 | 0.601 |
| Accumulator | 53 | 0.052 | 54 | 0.057 |

**Table 2.** Decision time difference statistics.

| | Depolarizing | | Hyperpolarizing | |
|---|---|---|---|---|
| | $B_1$ | p | $B_1$ | p |
| Total task-related input firing rates = 60 Hz | 50.442 | 0.043 | −56.366 | 0.037 |
| Refresh rate = 30 Hz | 90.272 | 0.024 | −83.599 | 0.036 |
| Refresh rate = 120 Hz | 93.289 | 0.015 | −90.707 | 0.03 |
| Inhibitory interneuron stimulation | 106.958 | 0.027 | 16.913 | 0.704 |
| Pyramidal cell stimulation only | 87.496 | 0.028 | −77.929 | 0.045 |
| Uniform stimulation | 60.71 | 0.157 | 56.522 | 0.168 |
| Reinitialization | 72.805 | 0.037 | −83.953 | 0.035 |
| Accumulator | 108.859 | <0.001 | −44.87 | 0.008 |

firing rates, allowing unbiased competition on the following trial. The simulations described above used an ISI of 2 s (matching the human experiment), which results in trials often beginning before residual activity from the previous trial has completely decayed (*Figure 2D*). In additional control stimulations using a range of ISIs, choice hysteresis behavior indeed decreased as the time between stimuli increased, as evidenced by shifts toward zero in the mean indecision point (main effect of ISI: F(3,76) = 14.439, p<0.001; *Figure 4A*) and the influence of the previous choice on the current decision (main effect of ISI: F(3,76) = 24.196, p<0.001; *Figure 4B*).

## Control simulations: Accumulator with independent interneuron pools

Two mechanisms determine behavior in competitive attractor network models: recurrent excitation within each pyramidal population and mutual inhibition between these populations via a common pool of inhibitory interneurons. Pure accumulator models such as the drift diffusion model are an alternate class of decision making models that do not include mutual inhibition (*Ratcliff and McKoon, 2008*, *1998*; *Ratcliff, 1978*). In these models, separate units integrate their inputs representing evidence for the corresponding option, and a decision is made when one unit reaches a predefined threshold. We tested whether separate integrators would make the same choice hysteresis predictions as the competitive attractor model. We split the interneuron population into two subpopulations, each exclusively connected with the corresponding pyramidal population (*Figure 5A*). The pyramidal populations could thus integrate their inputs through their recurrent excitatory connections, but could not exert any inhibitory influence on each other. All other parameters were kept the same, except for the background input firing rate and response threshold, as the resulting network

**Table 3.** Choice hysteresis simulation statistics.

| | Indecision point shifts (Left*-Right*) | | | | Logistic regression (a2/a1) | | | |
|---|---|---|---|---|---|---|---|---|
| | Depolarizing | | Hyperpolarizing | | Depolarizing | | Hyperpolarizing | |
| | W(19) | p | W(19) | p | W(19) | p | W(19) | p |
| Total task-related input firing rates = 60 Hz | 42 | 0.019 | 42 | 0.019 | 50 | 0.04 | 50 | 0.04 |
| Refresh rate = 30 Hz | 49 | 0.037 | 32 | 0.006 | 46 | 0.028 | 52 | 0.048 |
| Refresh rate = 120 Hz | 38 | 0.013 | 42 | 0.019 | 44 | 0.023 | 31 | 0.006 |
| Inhibitory interneuron stimulation only | 65 | 0.135 | 80 | 0.351 | 40 | 0.015 | 62 | 0.108 |
| Pyramidal cell stimulation only | 48 | 0.033 | 44 | 0.023 | 48 | 0.033 | 34 | 0.008 |
| Uniform stimulation | 79 | 0.332 | 43 | 0.021 | 99 | 0.823 | 73 | 0.232 |
| Reinitialization | 74 | 0.247 | 96 | 0.737 | 66 | 0.145 | 95 | 0.709 |
| Accumulator | 54 | 0.057 | 95 | 0.709 | 71 | 0.204 | 85 | 0.455 |

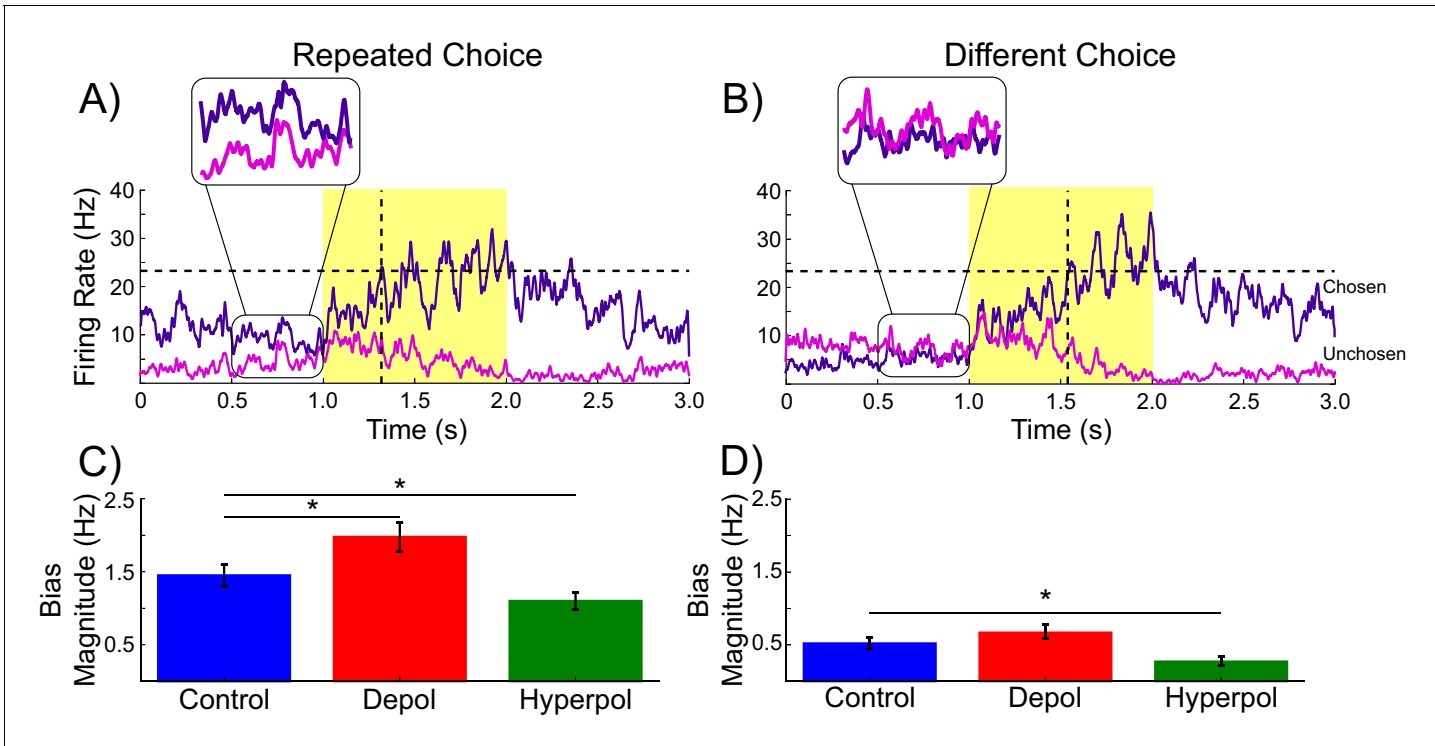

**Figure 3.** Decaying tail activity causes neural hysteresis effects. (**A**) Example trial in which the previous choice was repeated, showing a marked difference in the decaying tail activity from the previous trial between the eventually chosen and unchosen pyramidal populations, prior to the onset of the stimulus. (**B**) Example trial in which the pre-stimulus difference between chosen and unchosen firing rates is small. As a consequence, the bias in activity at stimulus onset is not strong enough to influence the choice. (**C**) The mean difference in firing rates in the 500 ms prior to the onset of the task-related inputs (magnified region in A) is amplified by depolarizing (red) and suppressed by hyperpolarizing (green) stimulation, relative to no stimulation (blue). (**D**) When the bias in pre-stimulus activity is relatively small, the model behavior is dominated by the task-related inputs, and therefore the model is able to overcome the hysteresis bias and make a different choice. See *Figure 3—source data 1* for raw data.

The following source data is available for figure 3:

**Source data 1.** Model prestimulus firing rates in repeated and non-repeated choice trials.

became more sensitive to these values (see Materials and methods). The accumulator version of the model made the same qualitative predictions as the competitive attractor version concerning accuracy and decision time (*Figure 5B–D*): choice accuracy is not affected by depolarizing (W(19) = 53, p=0.052) or hyperpolarizing stimulation (W(19) = 54, p=0.057), but depolarizing and hyperpolarizing stimulation speeds and slows decision time, respectively, and these effects are reduced with increasing coherence (depolarizing: $B_1$ = 108.859, p<0.001; hyperpolarizing: $B_1$ = −44.87, p=0.008). However, the model did not exhibit significant choice hysteresis (indecision point shift: W(19) = 77, p=0.296; $a_2/a_1$: W(19) = 86, p=0.478; *Figure 5E,F*). This is because the chosen pyramidal population does not inhibit the other population, allowing decaying trace activity from both populations to extend into the next trial (*Figure 5G*). Therefore, on the next trial both populations can be similarly biased. Neither depolarizing (indecision point shift: W(19) = 54, p=0.057; $a_2/a_1$: W(19) = 71, p=0.204) nor hyperpolarizing stimulation (indecision point shift: W(19) = 95, p=0.709; $a_2/a_1$: W(19) = 85, p=0.455) had any effect on choice hysteresis.

## Stimulation over human dlPFC directionally influences choice hysteresis in the same way as stimulation of a competitive attractor network

We next asked whether the predictions from our simulated stimulation were borne out in the behavior of human participants undergoing tDCS over dlPFC, a region strongly implied in controlling

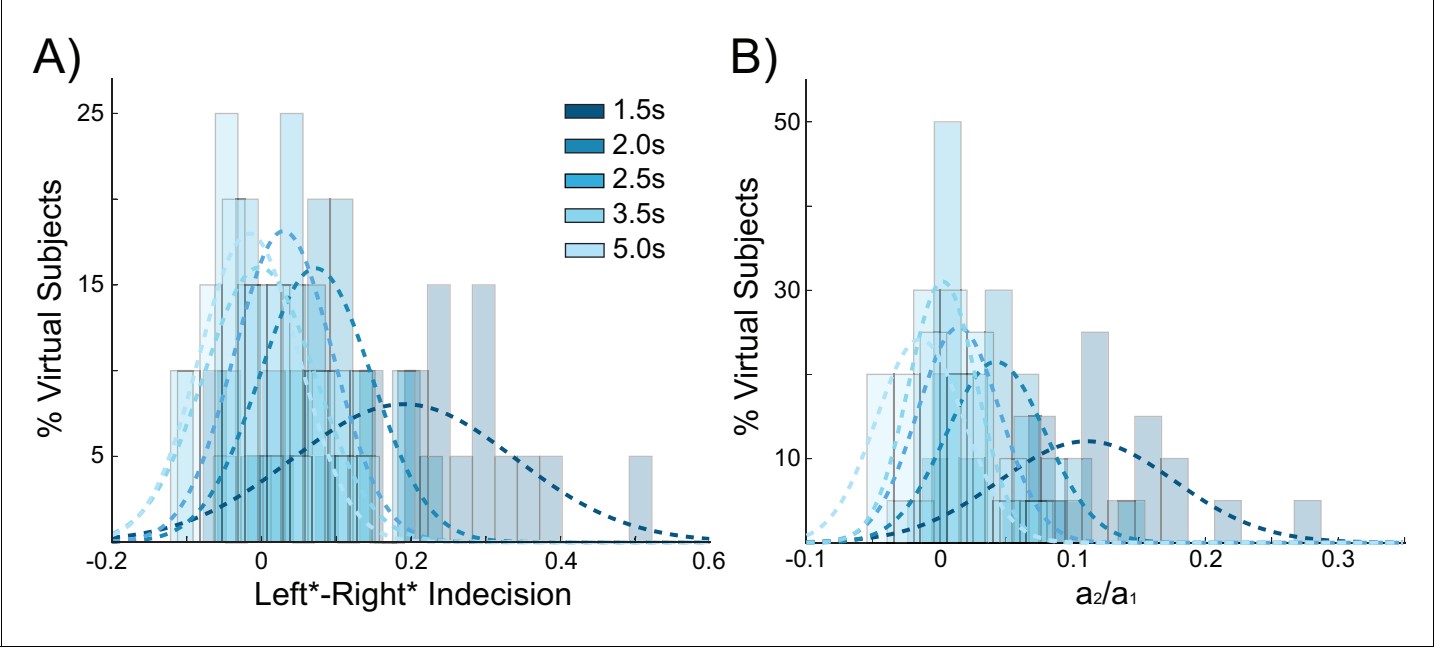

**Figure 4.** Behavioral choice hysteresis diminishes with longer interstimulus intervals. (**A**) The mean of the indecision point shift decreases with longer interstimulus intervals (ISIs), reflecting a smaller choice hysteresis effect. (**B**) Similarly, the ratio $a_1/a_2$ from the logistic regression, representing the relative influence of the previous choice on the current choice relative to the influence of coherence, decreases with increasing ISIs. Choice hysteresis is strongest for short ISIs of 1.5 s, and disappears for the longest ISI of 5 s. See *Figure 4—source data 1* for raw data.

The following source data is available for figure 4:

**Source data 1.** Model choice hysteresis behavior with increasing ISIs.

human perceptual choice (*Heekeren et al., 2004*, *2006*; *Philiastides et al., 2011*; *Rahnev et al., 2016*; *Georgiev et al., 2016*).

24 human participants performed the perceptual decision making task simulated in the model, in which they viewed a RDK and were required to indicate the direction of coherent motion (*Figure 6A*). To test the model's predictions concerning the effects of perturbed dynamics on choice hysteresis, we applied tDCS over the left dlPFC in order to induce depolarizing or hyperpolarizing network stimulation, or sham stimulation (*Figure 6B,C*). This region is implicated in perceptual decision making, independent of stimulus and response modality (*Heekeren et al., 2004*, *2006*), and it has been suggested that it operates using competitive attractor networks similar to the one we used (*Bonaiuto and Arbib, 2014*; *Rolls et al., 2010*; *Compte et al., 2000*; *Wimmer et al., 2014*). Furthermore, transcranial magnetic stimulation (TMS) over this region disrupts perceptual decisions, suggesting that it plays a necessary role in this process (*Philiastides et al., 2011*; *Rahnev et al., 2016*; *Georgiev et al., 2016*). However, here we employed tDCS, which subtly polarizes membrane potentials through externally applied electrical currents (*Rahman et al., 2013*; *Nitsche and Paulus, 2011*; *Bikson et al., 2004*), instead of disrupting ongoing neural activity, as with TMS. This allowed us to alter the dynamics of the human dlPFC in an analogous way to the model simulations.

As expected (*Palmer et al., 2005*), in sham stimulation blocks, the accuracy of human participants increased (*Figure 6D,E*) and response times decreased (*Figure 6F,G*) with increasing motion coherence. In striking accordance with the predictions of our biophysical model, neither depolarizing nor hyperpolarizing stimulation had an effect on the accuracy threshold of human participants (depolarizing: $W(23) = 145$, $p=0.886$; hyperpolarizing: $W(23) = 118$, $p=0.361$; *Figure 6D,E*). However, tDCS over left dlPFC in our human participants showed the predicted pattern of effects on response time for depolarizing and hyperpolarizing stimulation, compared to sham stimulation (depolarizing: $B_1 = 66.938$, $p=0.004$; hyperpolarizing: $B_1 = -49.677$, $p=0.04$; *Figure 6F–H*).

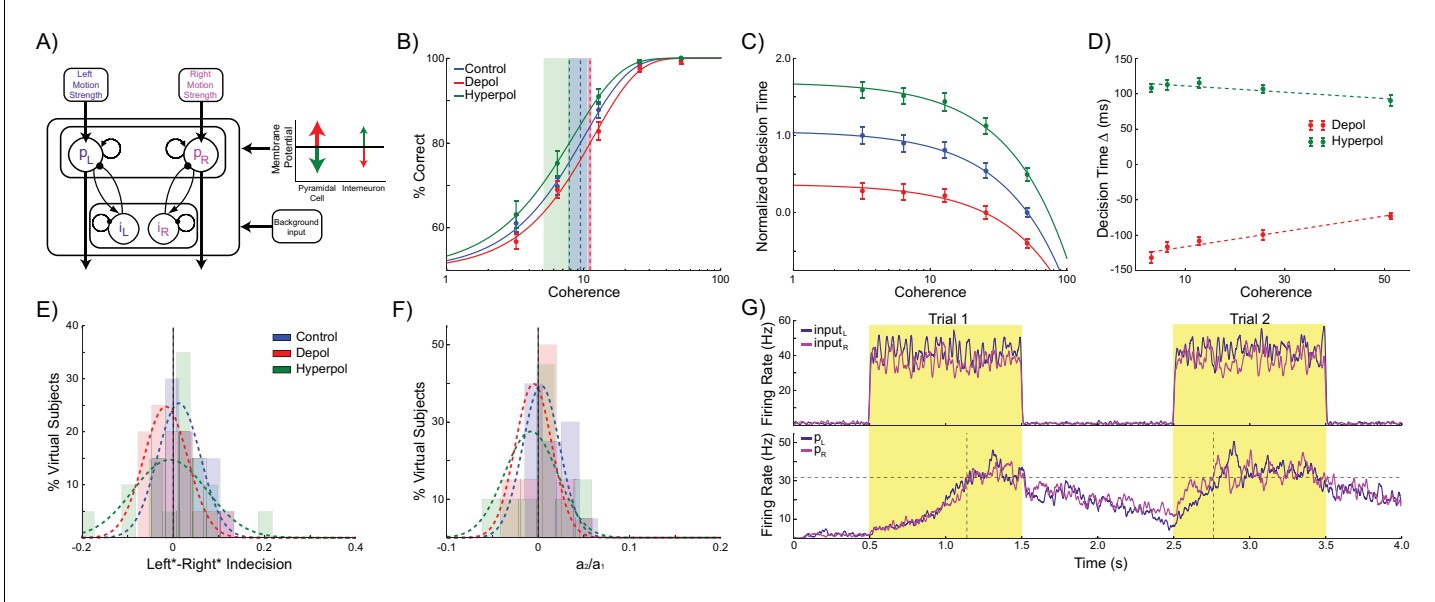

**Figure 5.** Accumulator model architecture and simulations. (**A**) In the accumulator version of the model, the inhibitory interneuron population is split into two subpopulations, each connected exclusively to the corresponding pyramidal population. The pyramidal populations can thus only integrate the task related inputs through their recurrent connectivity and cannot inhibit each other. (**B**) Neither depolarizing nor hyperpolarizing stimulation significantly changed the decision threshold. (**C**) As with the competitive attractor model, decision time decreases with increasing coherence, and depolarizing stimulation speeds decision time, while hyperpolarizing stimulation slows decisions. (**D**) The effects of stimulation on decision time are reduced with increasing coherence. (**E**) There is no shift in indecision point when sorting trials based on the previous choice, and stimulation does not affect this. (**F**) The relative influence of the previous choice on the current decision is nearly zero, and this is not changed by stimulation. (**G**). Neural dynamics of the accumulator model. The losing pyramidal population is not inhibited and therefore residual activity from both populations carries over into the next trial. See *Figure 5—source data 1* for raw data.

The following source data is available for figure 5:

**Source data 1.** Accumulator model accuracy, decision time, and choice hysteresis with simulated network stimulation.

The behavior of human participants additionally demonstrated choice hysteresis effects. Just as predicted by the model and in line with experimental work with humans and nonhuman primates (*Padoa-Schioppa, 2013*; *Samuelson and Zeckhauser, 1988*), the indecision points of human participants shifted when sorting their trials according to the preceding choice, and a logistic regression revealed a significant influence of the previous choice relative to coherence on decisions. We performed the same analyses used to examine choice hysteresis in the model on behavioral data from the human participants, now comparing each stimulation block to the preceding sham block in the same session. Exactly as predicted by the model, a shift in indecision point, which indicates a choice hysteresis effect, was observed under sham stimulation (depolarizing sham: W(23) = 79, p=0.043; hyperpolarizing sham: W(23) = 47, p=0.003), and this effect was amplified by depolarizing stimulation (W(23) = 78, p=0.04; *Figure 7A*) and reduced by hyperpolarizing stimulation (W(23) = 59, p=0.009; *Figure 7B*), relative to the preceding sham block. The results of the logistic regression confirmed these results, with a nonzero ratio of the influence of the previous choice to that of the current coherence in the sham stimulation blocks (depolarizing sham: W(23) = 73, p=0.028; hyperpolarizing sham: W(23) = 47, p=0.003), which was increased by depolarizing (W(23) = 56, p=0.007; *Figure 7D*) and decreased by hyperpolarizing stimulation (W(23) = 78, p=0.04; *Figure 7E*) relative to the preceding sham block. We therefore found that in both the model and in human participants, polarization of the dlPFC led to changes in reaction times and choice hysteresis effects in a perceptual decision making task. The neural dynamics of the model suggest that these changes are explained by alterations of sustained recurrent activity following a trial, which biases the decision process in the next trial.

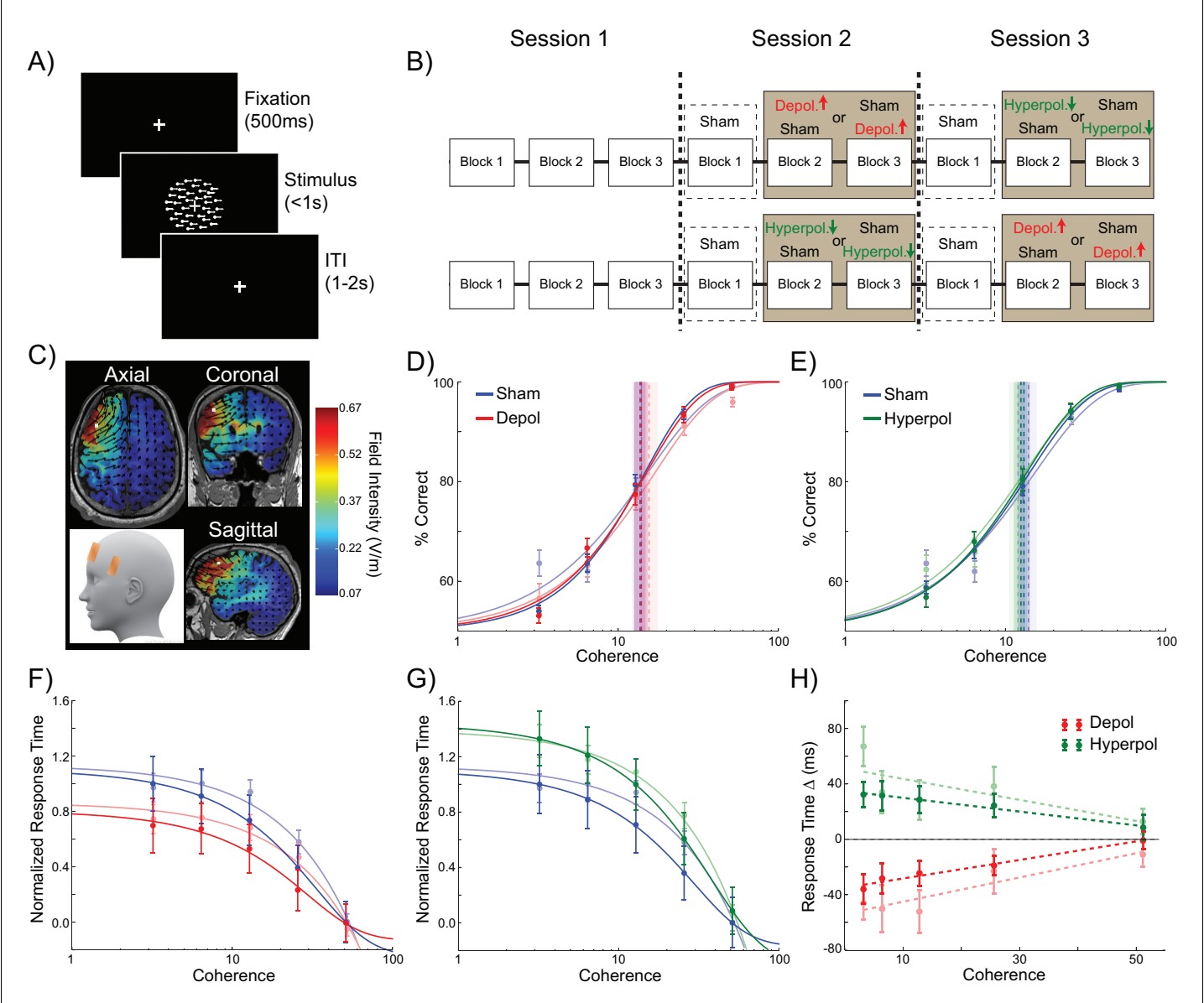

**Figure 6.** Experiment design and behavioral effects of stimulation in human participants. (A) Behavioral task. (B) Experimental protocol. (C) Electrode positions and estimated current distribution during left dorsolateral prefrontal stimulation. (D–E) Neither depolarizing nor hyperpolarizing stimulation altered the decision thresholds of human participants (model predictions shown in shaded colors in the background). (F–H) In human participants depolarizing and hyperpolarizing stimulation significantly decreased and increased the decision time, respectively, relative to sham stimulation, matching the model predictions shown in matte colors. See *Figure 6—source data 1* for raw data.

The following source data is available for figure 6:

**Source data 1.** Human participant accuracy and reaction time.

We next sought to further test the predictions of our model concerning the effect of ISIs on choice hysteresis. A separate group of participants (N = 24), performed a version of the task without stimulation in which the ISI was either 1.5 s or 5 s (see Materials and methods). Exactly as predicted by the model, trials following short ISIs had larger indecision point shifts (W(22) = 67, p=0.031; *Figure 8A*) and were more influenced by the previous choice (W(22) = 51, p=0.008; *Figure 8C*) compared to trials following longer ISIs. In fact with an ISI of 5 s, the indecision point was not significantly shifted (W(22) = 82, p=0.089), nor was there a detectable influence of the previous choice on the

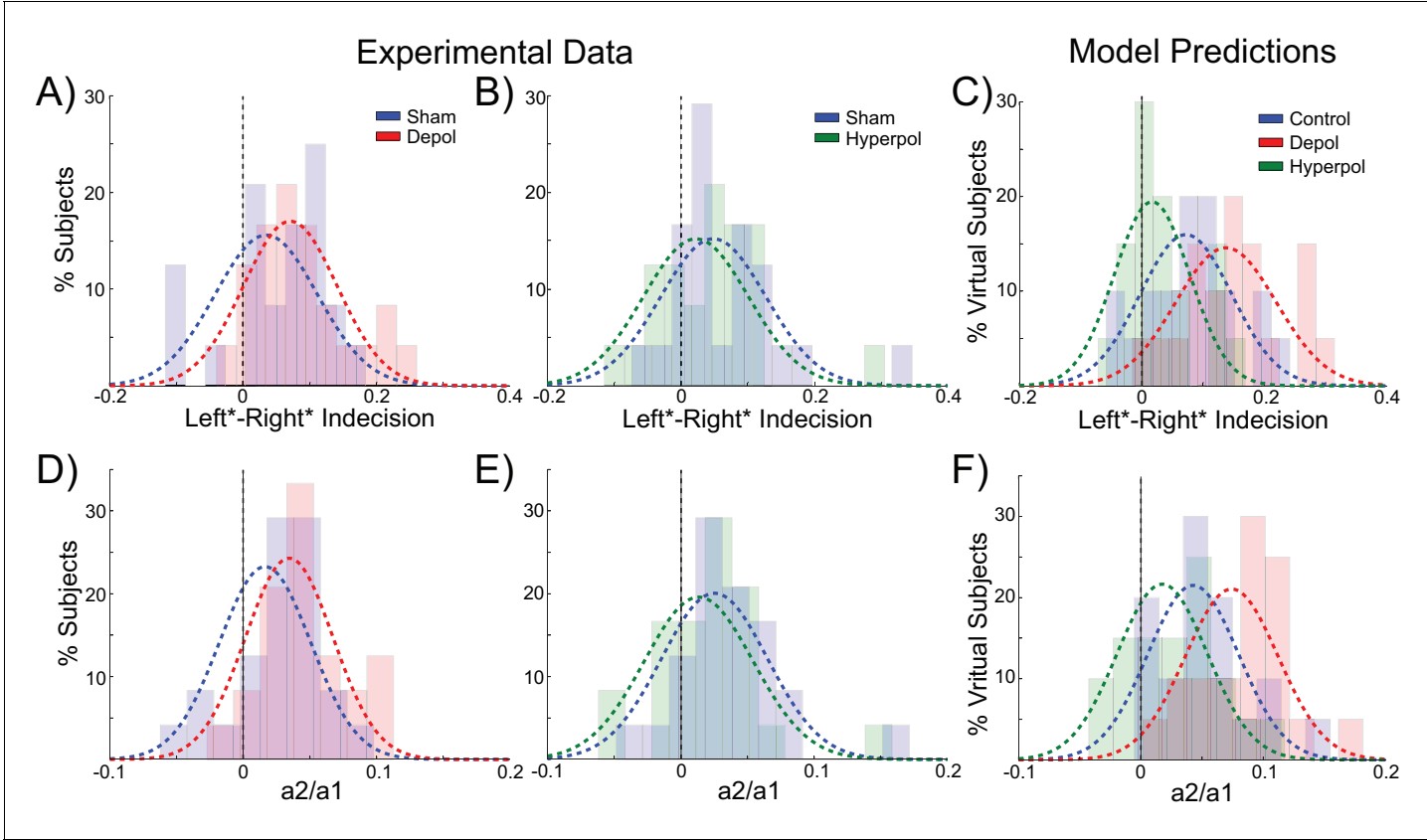

**Figure 7.** Behavioral effects of stimulation on choice hysteresis. (A–B) Depolarizing stimulation positively shifted the indecision point in human participants, while hyperpolarization caused the opposite effects, relative to sham stimulation. (C) These results are in line with model predictions (repeated from *Figure 2F*). (D–E) The relative influence of the previous choice in the current decision of human participants was increased by depolarizing and decreased by hyperpolarizing stimulation, relative to sham. (F) These results are just as predicted by the model (repeated from *Figure 2H*). See *Figure 7—source data 1* for raw data.

The following source data is available for figure 7:

**Source data 1.** Human participant behavioral choice hysteresis.

current decision (W(22) = 84, p=0.101), meaning that choice hysteresis had disappeared. The model explains this effect by the decay rate of the sustained recurrent activity in the winning pyramidal population, which has sufficient time to fall to baseline levels with long ISIs.

The indecision point shifts and logistic parameter ratios were not significantly different between the sham conditions of the stimulation experiment and the 1.5 s ISI condition (Mann-Whitney U test, indecision point shift, depolarizing sham: U = 253, p=0.632, indecision point shift, hyperpolarizing sham: U = 259, p=0.726, previous choice influence, depolarizing sham: U = 237, p=0.413, previous choice influence, hyperpolarizing sham: U = 254, p=0.647). In the stimulation experiment, the overall mean RT in sham conditions was 586 ms, resulting in a mean ISI of 1.914 s. It is therefore likely not different enough from the 1.5 s ISI condition to judge a significant difference between participant groups with our sample size.

## Discussion

Perceptual and economic decision making have been proposed to involve competitive dynamics in neural circuits containing populations of pyramidal cells tuned to each response option (*Wang, 2008*, *2012*, *2002*). Recent work has started to illuminate possible mechanisms for choice hysteresis biases and their neural correlates, but the putative neural mechanisms and dynamics underlying these

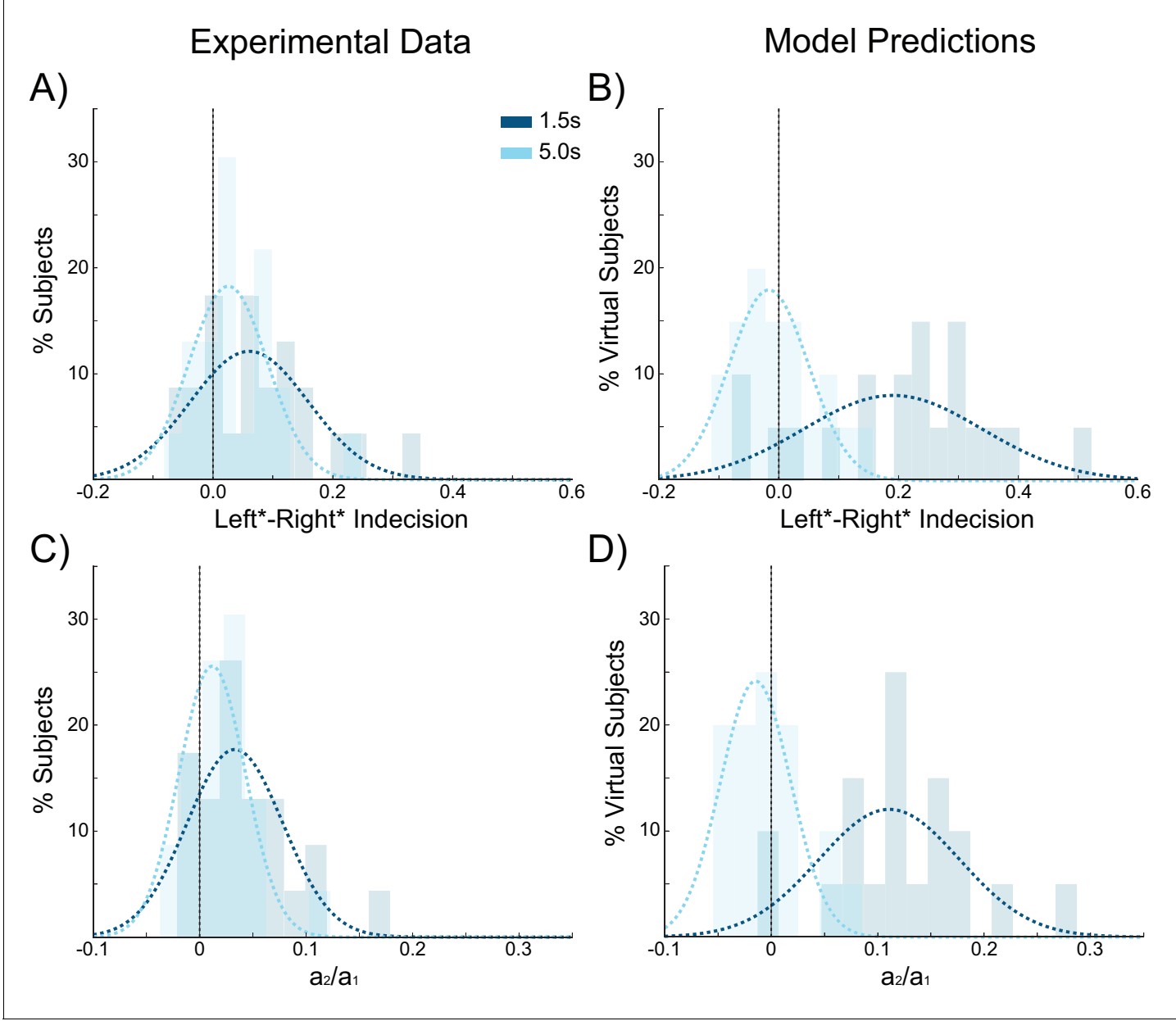

**Figure 8.** Choice hysteresis in human participants diminishes with longer interstimulus intervals. (A) The mean of the indecision point in human participants approaches zero with a 5 s interstimulus intervals (ISIs), reflecting a smaller choice hysteresis effect. (B) These results are similar to those predicted by the model (repeated from *Figure 4A*). (C) The relative influence of the previous choice on the current decision is smaller with a 5 s ISI than a 1.5 s ISI. (D) These results match the predictions of the model (repeated from *Figure 4B*). See *Figure 8—source data 1* for raw data.

The following source data is available for figure 8:

**Source data 1.** Human participant behavioral choice hysteresis with increasing ISIs.

biases have not been investigated with a causal approach in humans. We here used tDCS, a noninvasive neurostimulation technique, over left dlPFC to provide subtle perturbations of neural network dynamics while participants performed a perceptual decision making task. We leveraged recent developments in computational modeling approaches that bridge between the physiological and behavioral consequences of tDCS (*Hämmerer et al., 2016*; *Bonaiuto and Bestmann, 2015*; *Fröhlich, 2015*; *Bestmann, 2015*; *de Berker et al., 2013*; *Ruzzoli et al., 2010*; *Neggers et al., 2015*;

*Hartwigsen et al., 2015*; *Bestmann et al., 2015*) to generate behavioral predictions about how the gentle perturbation of network dynamics might impact behavior. To this end, we simulated the impact of tDCS in an established biophysical attractor model of dlPFC function. We show that carry-over activity from previous trials biases the activity of the network in the current trial, thus introducing a tendency to repeat the previous choice when the current one is difficult. Stimulation modulates the rate at which this carry-over activity decays, thus amplifying or suppressing choice hysteresis in the same way we observed in healthy participants undergoing analogous stimulation over dlPFC.

Our results also provide interventional evidence for the role of left dlPFC in perceptual decision making, and support the mechanistic proposal that this region integrates and compares sensory evidence through competitive interactions between pyramidal cell populations which are selective for each response option. Our modeling results suggest that stimulation over this region changes the level of sustained recurrent activity, and that this change modulates both choice variability and decision time. A competing accumulator version of our model that included integration without competition did not exhibit choice hysteresis. More generally, our approach demonstrates that a linkage between computational modeling and noninvasive brain stimulation allows mechanistic accounts of brain function to be causally tested.

## Behavioral effects of stimulation in silico are matched by analogous stimulation over left dlPFC in healthy participants

Our neural network model displayed decaying tail activity from the previous trial, leading to a 'choice hysteresis' effect, or a tendency to repeat the last choice, especially during difficult trials. This effect diminished with longer ISIs which allowed the tail activity to fully decay. Simulation of depolarizing noninvasive brain stimulation in this model decreased decision time and amplified this choice hysteresis effect, while hyperpolarizing stimulation increased decision time and suppressed choice hysteresis. These behavioral effects were caused by changes in the level of sustained activity from the previous trial, placing the network closer or further away to one of its attractor states, biasing the response to one or the other option, thus speeding or slowing decisions. Human participants demonstrated the same choice hysteresis bias, which was diminished with longer ISIs (but see *Akaishi et al., 2014*) and modulated by noninvasive brain stimulation over the left dlPFC in the same way. This supports the idea that processes related to perceptual decisions in left dlPFC are underpinned by processes that can be approximated by a competitive attractor network as used here.

Although not statistically significant, simulated depolarizing stimulation slightly decreased choice accuracy, while hyperpolarizing stimulation improved it. This is not surprising, given the effects of stimulation on decision time and the use of a firing rate threshold as a decision criterion. However, in previous studies with a similar model we found that decision making accuracy is affected at higher levels of stimulation intensity (*Bonaiuto and Bestmann, 2015*). The stimulation intensities used in the present simulations were matched to those used in the human experiments, and not sufficiently high to polarize the network enough to completely override differences in task-related inputs (left/right motion coherence). Previous work has shown that repetitive TMS over left dlPFC reduces both accuracy and increases response time in a similar task (*Philiastides et al., 2011*). However, TMS elicits instantaneous synchronized activity within the area of stimulation, and thus likely disrupts ongoing activity, providing an 'override' of difference in task-related inputs. By contrast, tDCS subtly alters neural dynamics through small de- or hyper-polarizing currents without directly eliciting spikes (*Radman et al., 2009*; *Bikson et al., 2013*). With the number of trials used and the variability observed in the present task, it is likely that stimulation effects were only apparent in response time because response time is a more sensitive performance metric than choice accuracy. More generally, our results show that the possible mechanisms through which non-invasive brain stimulation alters behavior can be interrogated through the use of biologically informed computational modeling approaches. Computational neurostimulation of the kind employed here may thus provide an important development for mechanistically informed rationales for the application of neurostimulation in both health and disease.

## Study limitations

Here, we simulated the effects of both depolarizing and hyperpolarizing dlPFC stimulation. In these simulations, currents affected both pyramidal cells as well as interneurons. This modeling choice was based on previous work showing that simulated tDCS must affect both pyramidal cells and interneurons in order to explain changes in sensory evoked potentials observed in vitro (*Molaee-Ardekani et al., 2013*). Some accounts of the neurophysiological effects of tDCS suggest that pyramidal neurons are predominantly affected (*Radman et al., 2009*), but in additional simulations in which stimulation was only applied to the pyramidal cells the results were qualitatively similar (*Tables 1–3*).

The level of stimulation intensity and montages we used for our human experiments are based on current modeling estimates of the mean field strength in dlPFC and in vitro measurements of pyramidal cell and interneuron polarization as a function of field strength (*Bikson et al., 2004*). We also used individual structural MRIs to optimize electrode placement for each participant, using relatively small stimulation electrodes that straddled our target site in left dlPFC. Specifically, electrode positions relative to the MNI template were determined using current modeling to maximize current flow *through* the superior frontal sulcus portion of the left dlPFC, creating radial inward or outward current flow. Each participant's MRI was aligned to the template and the inverse of this transformation was used to derive optimal electrode positions to generate current flow through the same sulcus. This approach ensured that stimulation was relatively confined over left prefrontal cortex, and that the direction of current flow through the target site was comparable across subjects. However, in our simulations neurons were not spatially localized and polarization was applied uniformly to all neurons within a population. Future computational neurostimulation studies should investigate the effects of heterogeneous polarization due to variable patterns of current flow through brain tissue.

We targeted the left dlPFC, but perceptual decision making involves a network of cortical regions including the middle temporal area MT (*Salzman et al., 1992*; *Britten et al., 1993*; *Celebrini and Newsome, 1994*) and the lateral intraparietal area LIP (*Hanks et al., 2006*; *Shadlen and Newsome, 1996*). However, the left dlPFC is an attractive target for our aims for a number of reasons. The activity of neurons in dlPFC both predicts the upcoming response and reflects information about the sensory stimuli, suggesting that this region makes an integral contribution to transforming sensory information into a decision (*Kim and Shadlen, 1999*). Furthermore, in human perceptual decision making, left dlPFC is activated independent of the both the stimulus type and response modality (*Heekeren et al., 2004*, *2006*; *Pleger et al., 2006*; *Wenzlaff et al., 2011*; *Ruff et al., 2010*; *Philiastides and Sajda, 2007*; *Kovács et al., 2010*; *Donner et al., 2007*; *Ostwald et al., 2012*; *Zhang et al., 2013*), and has been shown to play a causal role in the process (*Philiastides et al., 2011*). We optimized the electrode locations to stimulate the left dlPFC, while leaving premotor and motor cortices unaffected, as stimulation of these regions could induce response biases. This may not have been possible with stimulation over parietal cortex. Finally, neural hysteresis in another frontal region, orbitofrontal cortex, has been linked to behavioral choice hysteresis in value-based decisions in nonhuman primates (*Padoa-Schioppa, 2013*). Recent work suggests that this region can indeed be targeted with tDCS (*Hämmerer et al., 2016*), and whether choice hysteresis for value-based decision can be similarly molded as in the present study remains to be seen. However, decaying trace activity in left dlPFC could partially reflect lingering activity in afferent regions, and this possibility could be addressed in future research using a multiregional computational neurostimulation approach.

Participants made all responses using the hand contralateral to the site of stimulation (i.e. the right hand). It is therefore not clear what effect ipsilateral stimulation would have. However, in non-human primates, neurons in dlPFC are active during perceptual decision making whether the choice is indicated with a button press (*Hussar and Pasternak, 2013*) or a saccade (*Kim and Shadlen, 1999*; *Kiani et al., 2014*; *Opris and Bruce, 2005*). In humans, left dlPFC is activated during perceptual decision regardless of the response modality (e.g., saccades versus button presses [*Heekeren et al., 2006*]), right-handed button presses (*Heekeren et al., 2006*; *Ruff et al., 2010*; *Philiastides and Sajda, 2007*; *Donner et al., 2007*), left-handed button presses (*Pleger et al., 2006*), and bimanual button presses (*Wenzlaff et al., 2011*). It is thus reasonable to expect that stimulation of the left dlPFC would affect both hands in the same way. Investigating the potential lateralization of stimulation-induced effects on dlPFC as well as other regions in the perceptual decision

making network such as MT and LIP, is an interesting avenue for further research for which the computational neurostimulation approach employed here could be leveraged.

We found no choice hysteresis bias in the accumulator version of the model, with or without simulated stimulation. However, compared to the competitive attractor model, the accumulator model made very similar predictions concerning choice accuracy and decision time, as well as the effects of stimulation on these measures. In our simulations, the lack of mutual inhibition in the accumulator model allows residual activity from both pyramidal populations to bias the following decision, resulting in zero net bias. In constructing the accumulator model, we sought to alter the competitive attractor model as little as possible, keeping most parameters at the same value. However, we found that without mutual inhibition, the firing rates of the resulting network were much more sensitive to the noisy background inputs, and therefore we had to restrict the range of background firing rates used and scale the response threshold accordingly. It is possible that there are some sets of parameter values that would cause the accumulator model to exhibit choice hysteresis, but a systematic search of the parameter space of this model is beyond the scope of this study.

While the model we used to simulate the dlPFC is well established, it is a general model used to study decision making. As such, it does not take into account the specific architecture and detailed connectivity of the dlPFC; however data at this level of detail in general are not currently available. The choice hysteresis behavior in human participants qualitatively matched that of the model, but the model predicted a larger effect than that observed (*Figures 7* and *8*). Factors such as the contribution of other brain regions known to be involved in perceptual decision making may explain these quantitative differences. Future studies should involve increasingly realistic biophysical multi-region models that can take this information into account.

## Computational neurostimulation

tDCS is widely used in basic and translational studies for reversible and controlled modulation of neural circuit activity in the human brain. However, there is a distinct lack of mechanistic models that not only explain how stimulation affects neural network dynamics, but also how these changes alter behavior. There are several conceptual models of the effects of noninvasive brain stimulation, but the explanations that they offer typically make leaps across several levels of brain organization and don't consider how neural circuits generate behavior (*de Berker et al., 2013*; *Bestmann et al., 2015*). Efforts have been made in this direction (*Hämmerer et al., 2016*; *Rahman et al., 2013*; *Bonaiuto and Bestmann, 2015*; *Molaee-Ardekani et al., 2013*; *Fröhlich, 2015*; *Bestmann, 2015*; *Neggers et al., 2015*; *Hartwigsen et al., 2015*; *Miniussi et al., 2013*; *Rahman et al., 2015*), but there have been very few computational models that offer an explanation for the behavioral effects of tDCS in terms of neural circuit dynamics (*Hämmerer et al., 2016*; *Bonaiuto and Bestmann, 2015*; *Douglas et al., 2015*). We here present the first biophysically informed modeling study of tDCS effects during perceptual decision making, and provide detailed hypotheses about the changes in neural circuits that translate to observed behavioral changes during stimulation.

## Conclusion

We have shown that a biophysical attractor model generates perceptual decision making behavior accurately matching that of human participants. Additionally, we used computational neurostimulation of this model to predict the effect of tDCS over left dlPFC on choice hysteresis, which we then confirmed experimentally. Previous work showing that changes to parameters in diffusion models can explain differences in human perceptual decision making after transcranial magnetic stimulation of left dlPFC (*Philiastides et al., 2011*; *Rahnev et al., 2016*; *Georgiev et al., 2016*). Our results extend these findings by addressing the neural circuitry behind perceptual decision making processes. This allows us to capture the influence of previous neural activity on the current choice in a natural way, and to offer mechanistic explanations for the effects of stimulation on neural dynamics in the left dlPFC. We provide interventional evidence that the left dlPFC integrates and compares perceptual information through competitive interactions between neural populations, and that decaying trace activity from previous trials influences the current choice by biasing the decision toward the repeating the last choice.

## Materials and methods

### Biophysical attractor model

The model contains 2000 neurons, and consists of one population of 1600 pyramidal cells and one population of 400 inhibitory interneurons. The pyramidal cells contain two subpopulations of 240 neurons each, selective for the 'left', $p_L$, and 'right', $p_R$, choice options, with the remaining neurons non-selective for either option. The neurons in the pyramidal subpopulations form excitatory reciprocal connections with neurons in the same population, and mutually inhibit each other indirectly via projections to and from a common pool of inhibitory interneurons. The neurons in the pyramidal population project to excitatory synapses (AMPA and NMDA) on target cells and the interneurons project to inhibitory synapses on their targets (GABA$_A$).

All neural populations receive stochastic background input from a common pool of Poisson spike generators, causing each neuron to spontaneously fire at a low rate. The $p_L$ and $p_R$ pyramidal subpopulations additionally receive task-related inputs as Poisson distributed random spikes signaling the perceived evidence for each response option using the same scheme as Wang (*Wang, 2002*). The mean rates of the task-related inputs, $\mu_L$ and $\mu_R$, vary linearly with the coherence level of the simulated RDK (*Figure 1A*, inset). Importantly, the sum of the mean task-related input rates always equals 80 Hz, meaning that decision making behavior has to emerge from network dynamics and input structure and cannot be attributed to differences in the overall level of task-related input stimulation. The firing rate of each task-related input at each time point was normally distributed around the mean (σ = 4 Hz) and changed according to the refresh rate of the monitor used in our experiment (*Figure 1B*, left column). In additional simulations we show that the behavior of the model is qualitatively robust to changes in the total task-related input firing rates and refresh rate (*Tables 1–3*).

For analysis, we compute mean population firing rates by convolving the instantaneous population firing rate with a Gaussian filter 5 ms wide at the tails. The winner-take-all dynamic of the network causes the firing rates of the pyramidal populations to magnify differences in the inputs. This is due to the reciprocal connectivity and structure of the network, which endows it with bistable attractor states, resulting in competitive dynamics (*Camperi and Wang, 1998*; *Wilson and Cowan, 1972*). As the firing rate of one population increases, it increasingly inhibits the other population via the common pool of inhibitory interneurons. This further increases the activity of the winning population as the inhibitory activity caused by the other population decreases. These competitive dynamics result in one pyramidal population (typically the one receiving the strongest input) firing at a relatively high rate, while the firing rate of the other population decreases to approximately 0 Hz (*Figure 1B*).

### Synapse and neuron model

Here we provide a detailed description of the architecture of the biophysical attractor model we used. We modeled synapses as exponential (AMPA, GABA$_A$) conductances, or bi-exponential conductances (NMDA). Synaptic conductances are governed by the following equation:

$$g(t) = Ge^{-t/\tau} \tag{1}$$

where $G$ is the maximal conductance (or weight) of that specific synapse type (AMPA or GABA$_A$), and $\tau$ is the decay time constant for that synapse type. Thus when a spike arrives at this synapse at time $t$, the conductance, $g$, is set to its maximal value, $G$ (because $e^{-t/\tau}$ is bounded by 0 and 1), after which it decays at a rate determined by $\tau$. Similarly, bi-exponential synaptic conductances are determined by:

$$g(t) = G\frac{\tau_2}{\tau_2 - \tau_1}\left(e^{-t/\tau_1} - e^{-t/\tau_2}\right) \tag{2}$$

where $\tau_1$ and $\tau_2$ are rise and decay time constants. Synaptic currents are computed from the product of these conductances and the difference between the membrane potential and the synaptic current reversal potential, $E$:

$$I(t) = g(t)(V_m - E) \tag{3}$$

where $V_m$ is the membrane voltage. NMDA synapses have an additional voltage dependence, which is captured by:

$$I_{NMDA}(t) = \frac{g_{NMDA}(t)(V_m - E_{NMDA})}{1 + [Mg^{2+}]exp(-0.062V_m)/3.57} \quad (4)$$

where $[Mg^{2+}]$ is the extracellular magnesium concentration.

The total synaptic current (summing AMPA, NMDA, and GABA$_A$ currents) is input into the exponential leaky integrate-and-fire (LIF) neural model (*Brette and Gerstner, 2005*):

$$I_{total}(t) = I_{AMPA}(t) + I_{NMDA}(t) + I_{GABA_A}(t) \quad (5)$$

$$C\frac{dV_m}{dt} = g_L(V_m - E_L) + g_L\Delta_T e^{\frac{V_m - V_T}{\Delta_T}} - I_{total} \quad (6)$$

where $C$ is the membrane capacitance, $g_L$ is the leak conductance, $E_L$ is the resting potential, $\Delta_T$ is the slope factor (which determines the sharpness of the voltage threshold), and $V_T$ is the threshold voltage. After spike generation, the membrane potential is reset to $V_r$ and the neuron cannot generate another spike until the refractory period, $\tau_r$, has passed. Intra- and inter-population connections are initialized probabilistically with axonal conductance delays of 0.5 ms. Parameter values are based on experimental data from the literature where possible (*Hestrin et al., 1990*; *Jahr and Stevens, 1990*; *Salin and Prince, 1996*; *Spruston et al., 1995*; *Xiang et al., 1998*) and set empirically otherwise (*Table 4*).

All connection probabilities are determined empirically so that the network generates winnertake-all dynamics (*Bonaiuto and Arbib, 2014*; *Bonaiuto and Bestmann, 2015*). Recurrent pyramidal population connectivity probability (the probability that any pyramidal cell projected to an AMPA or NMDA synapse on other cells in the same population) was 0.08, and recurrent inhibitory interneuron population connections used GABA$_A$ synapses and had a connectivity probability of 0.1. Projections from the pyramidal populations connected to AMPA or NMDA synapses on the inhibitory interneurons with probability 0.1, connections from the inhibitory interneuron population to each pyramidal population used GABA$_A$ synapses with a connectivity probability of 0.2. Thus, the pattern of connectivity between populations was fixed, but the fine-scale connectivity between individual neurons was probabilistically determined by the connectivity parameters.

## Simulation of tDCS-Induced currents

We simulated depolarizing tDCS by injecting a depolarizing transmembrane current into each pyramidal cell and hyperpolarizing current into each interneuron (*Bonaiuto and Bestmann, 2015*; *Molaee-Ardekani et al., 2013*), and hyperpolarizing tDCS by adding hyperpolarizing current into pyramidal cells and depolarizing current into interneurons, a distinction which arises in cortex due to differences in orientation and cellular morphology. Note however that the results of our model are robust against these assumptions, and remain qualitatively similar when omitting current from interneurons (see below). The simulated current was added to the input to each exponential LIF neuron:

$$I_{total}(t) = I_{AMPA}(t) + I_{NMDA}(t) + I_{GABA_A}(t) + I_{stim}(t) \quad (7)$$

where $I_{stim}(t)$ is the tDCS current at time $t$.

The simulated tDCS current was applied for the entire duration of each block of trials. Depolarizing tDCS was simulated by injecting 0.75 pA into pyramidal cells and −0.375 pA into interneurons, while during hyperpolarizing tDCS stimulation pyramidal cells were injected with −0.75 pA and interneurons 0.375 pA (*Bonaiuto and Bestmann, 2015*; *Molaee-Ardekani et al., 2013*). In additional simulations when we applied stimulation only to the pyramidal populations the results were qualitatively similar, however when we applied stimulation only to the interneuron population or uniform stimulation to both populations the results were very different (see *Tables 1–3* and Discussion, [*Bestmann et al., 2015*; *Bonaiuto and Bestmann, 2015*]). The latter two stimulation protocols are in contrast with known physiology of polarizing currents (*Rahman et al., 2013*; *Radman et al., 2009*) and thus served as additional tests for the specificity of our simulated membrane polarization effects on the model. The injected current simulating tDCS slightly changed the resting membrane potential

**Table 4.** Parameter values for the competitive attractor model.

| Parameter | Description | Value |
|---|---|---|
| $G_{AMPA(ext)}$ | Maximum conductance of AMPA synapses from task-related inputs | 1.6nS |
| $G_{AMPA(background)}$ | Maximum conductance of AMPA synapses from background inputs | 2.1nS (pyramidal cells), 1.53nS (interneurons) |
| $G_{AMPA(rec)}$ | Maximum conductance of AMPA synapses from recurrent inputs | 0.05nS (pyramidal cells), 0.04nS (interneurons) |
| $G_{NMDA}$ | Maximum conductance of NMDA synapses | 0.145nS (pyramidal cells), 0.13nS (interneurons) |
| $G_{GABA-A}$ | Maximum conductance of GABA$_A$ synapses | 1.3nS (pyramidal cells), 1.0nS (interneurons) |
| $\tau_{AMPA}$ | Decay time constant of AMPA synaptic conductance | 2 ms |
| $\tau_{1-NMDA}$ | Rise time constant of NMDA synaptic conductance | 2 ms |
| $\tau_{2-NMDA}$ | Decay time constant of NMDA synaptic conductance | 100 ms |
| $\tau_{GABA-A}$ | Decay time constant of GABA$_A$ synaptic conductance | 5 ms |
| $[Mg^{2+}]$ | Extracellular magnesium concentration | 1 mM |
| $E_{AMPA}$ | Reversal potential of AMPA-induced currents | 0 mV |
| $E_{NMDA}$ | Reversal potential of NMDA-induced currents | 0 mV |
| $E_{GABA-A}$ | Reversal potential of GABA$_A$-induced currents | −70 mV |
| $C$ | Membrane capacitance | 0.5nF (pyramidal cells), 0.2nF (interneurons) |
| $g_L$ | Leak conductance | 25nS (pyramidal cells), 20nS (interneurons) |
| $E_L$ | Resting potential | −70 mV |
| $\Delta_T$ | Slope factor | 3 mV |
| $V_T$ | Voltage threshold | −55 mV |
| $V_s$ | Spike threshold | −20 mV |
| $V_r$ | Voltage reset | −53 mV |
| $\tau_r$ | Refractory period | 2 ms (pyramidal cells), 1 ms (interneurons) |

of each neuron (±0.038 mV with ±0.75 pA injected current, ±0.019 mV with ±0.375 pA), within the range found by in vitro tDCS studies (*Rahman et al., 2013*; *Bikson et al., 2004*; *Radman et al., 2009*).

## Simulation of the perceptual decision making task

One advantage of computational modeling is that the model can be run for many more trials than can feasibly be tested in human participants. However, this can lead to spuriously low-variance model predictions that cannot reliably be compared with human data. In order to fairly compare model and human behavioral performance, we generated 20 virtual subjects and assessed the effects of stimulation in each subject. Virtual subjects were generated using a random seed to generate fine grained neuron-to-neuron connectivity using the connection probabilities described above. Each virtual subject had a background input firing rate sampled from a range (880–950 Hz) previously used to simulate human participants in a similar decision making task (*Bonaiuto and Bestmann, 2015*) and a response threshold uniformly sampled from a range of 18–22 Hz to capture inter-subject differences in speed-accuracy tradeoffs. Simulations with the accumulator version of the model used a background rate between 855 Hz and 870 Hz and a response threshold that varied with the rate in the range 19–36 Hz. This was necessary because without mutual inhibition, the resulting network was much less stable and more sensitive to these values.

Each virtual subject was tested using the same five coherence levels that human participants were tested with (3.2, 6.4, 12.8, 25.6, and 51.2%), with 20 trials at each level (10 trials with coherent

motion to the left and 10 to the right), for a total of 100 trials per block, randomly ordered. The sum of the two task-related inputs always equaled 80 Hz (at coherence = 0% both inputs were at 40 Hz, and coherence = 51.2% one input was at 60.48 Hz and the other at 19.52 Hz), so the total strength of the input received by the network remains equal across all conditions. Each trial lasted for 3s, with task-related input applied from 1–2s, matching the average time course of the experiment with human participants. Three blocks of trials were run: no stimulation, depolarizing, and hyperpolarizing stimulation. In both stimulation blocks, stimulation was applied for the entire duration of the block: thus matching the experimental procedure in humans where we applied tDCS in a blocked manner.

All of our model simulations were implemented in the Python programming language using the Brian simulator v1.4.1 (*Goodman and Brette, 2008*). The differential equations defining the model were solved using Euler integration with a time step of 0.5 ms. Model simulation and analysis code is available at https://github.com/jbonaiuto/perceptual-choice-hysteresis.

## Human participants

24 neurologically healthy volunteers participated in the stimulation experiment (seven male, aged 23.75 ± 4.25 years), and a separate group of 24 participated in a control experiment assessing the influence of ISI (nine male, aged 23.54 ± 3.32 years). One of the participants in the ISI experiment was excluded from analysis because of their high accuracy threshold (>25%). The required number of participants was determined based on a power analysis of with an alpha of 0.05, power of 0.8 and effect sizes estimated from previous tDCS studies targeting dlPFC ($d$ = 0.6–0.9, [*Boggio et al., 2010*; *Fecteau et al., 2007*; *Jo et al., 2009*; *Fregni et al., 2005*]). Participants gave their informed written consent before participating and the local ethics committee approved the experiments (reference number 5833/001).

## Behavioral task

Participants completed a perceptual decision making task. Participants sat comfortably at a desk in front of a computer and responded to visual stimuli displayed on a screen by pressing two keys on a keyboard using the index and middle finger of their right hand. The screen had an update rate of 60 Hz and was placed 76 cm from the participants. On each trial, participants were required to fixate in the center of a screen. After 500 ms a RDK was displayed and participants were required to press a key as soon as possible to indicate whether the direction of coherent motion was to the left or the right (*Figure 6A*). Although only one direction of coherent motion was displayed during each trial, the task is a two alternative forced choice task, and therefore evidence must be accumulated and a decision made between the left and right direction. The RDK consisted of a 5° diameter circular aperture centered on the fixation point (*Ruzzoli et al., 2010*) with 0.1° diameter dots at a density of 16.7 dots/deg$^2$/s (*Britten et al., 1992*), each moving at 5°/s (*McGovern et al., 2012*). The percentage of coherently moving dots was set randomly in each trial to 3.2, 6.4, 12.8, 25.6, or 51.2%. Trials ended once a response had been made or after a maximum of 1s if no response was made. The inter-trial interval was 1–2s and varied depending on the response time of the previous trial to make all trials the same length. Combined with the 500 ms fixation period, ISIs were therefore between 1.5 and 2.5s. Matching the model simulations, each block contained 10 trials for each coherence level with half containing coherent leftward motion and half rightward (100 trials total). All trials were randomly ordered. Participants were shown cumulative feedback at the end of each block displaying % correct, the mean response time in the most difficult trials, and # correct responses / minute. Before each session, participants completed a training block in which trial-by-trial feedback was given during the first ten trials. We used a within-subject design in which each participant completed three sessions (depolarizing stimulation; hyperpolarizing stimulation; no stimulation; *Figure 6B*). The order of the stimulation conditions was balanced across participants. The human behavioral task was implemented in Python using PsychoPy v1.78.01 (*Peirce, 2007*).

## Transcranial direct current stimulation

Transcranial Direct Current Stimulation (tDCS) was applied over the left dorsolateral prefrontal cortex (dlPFC; *Figure 6C*) using a battery-driven multi-channel direct current stimulator (NeuroConn, GmbH). Specifically, activation within the lateral wall of superior frontal sulcus in posterior left dlPFC relates to perceptual decision making regardless of the response modality ([x, y, z,=-23, 29, 37],

[*Heekeren et al., 2004*, *2006*; *Ostwald et al., 2012*; *Zhang et al., 2013*]). We optimized electrode positions for targeting of the left dlPFC using MRI-derived head models of electric field (EF) distributions to maximize current flow through this voxel (HD-Explore and HD-Targets software, v4.0, Soterix Medical, New York, NY, USA). We determined that electrode positions on the scalp approximately 5 cm medial and lateral to the nearest point to the dlPFC voxel maximized current flow through the lateral sulcal wall of the target location (*Figure 6C*), whilst sparing premotor/motor cortices. Inward/anodal (relative to the cortical sulcal surface) currents have an opposite effect on neural polarization to outward/cathodal currents (*Rahman et al., 2013*; *Bonaiuto and Bestmann, 2015*; *Bestmann et al., 2015*). Placing the cathode electrode in the medial position and the anodal electrode in the lateral position maximized outward (cathodal) current, while the opposite configuration maximized inward current (anodal) flow through the target site (*Rahman et al., 2013*; *Bikson et al., 2004*; *Radman et al., 2009*; *Basser and Roth, 2000*; *Reato et al., 2010*).

Individualized electrode positions for each participant were derived using their structural MRI scan. Each participant's MRI was aligned to the MNI template and the dlPFC coordinate was localized in native space using the inverse co-registration transformation. The coordinate was then used in the neuronavigation software (Visor) to mark the nearest point in the superior frontal sulcus, and from this an electrode location on the forehead corresponding to a location on the scalp radial from this target site.

Participants completed a total of three sessions spaced approximately one week apart (*Figure 6B*). In the first session, participants completed three blocks of 100 trials each with three short breaks. Each block lasted 20 min. During each session with stimulation (depolarizing or hyperpolarizing with the session order balanced across participants), participants completed two blocks of trials with sham stimulation, and one with depolarizing or hyperpolarizing stimulation. The first block was always sham and the order of the second and third blocks was balanced across participants and stimulation conditions. This within-subjects design was chosen to maximize statistical power and control for learning effects, as we found in a pilot study that performance on the task improved between sessions as well as between blocks within a session. Behavior from each stimulation block was compared with the sham block directly preceding it in the same session, controlling for both within- and between-session learning. During stimulation blocks, tDCS was applied for 20 min at 2 mA. During sham blocks the stimulation was ramped up to 2mA over 10 s, stimulated for 30 s, and then ramped down to 0 over 10 s.

## Assessing the impact of ISI on choice hysteresis

The model predicted that choice biases should diminish with longer ISIs, because of the longer time for neural activity to return to baseline. To test this model prediction, we conducted a control experiment using the same task as that in the main experiment, with the exception that the inter-trial intervals were either 1 or 4.5 s long. Combined with the 500 ms fixation duration, this resulted in ISIs of either 1.5 or 5 s and these were randomized within blocks. Following a training block in which trial-by-trial feedback was given, participants completed three test blocks without feedback. Each test block contained 20 trials for each combination of ISI and coherence level, with half containing coherent leftward motion and half rightward (200 trials total per block). All trials were randomly ordered. The analysis was performed on all test blocks.

## Data analyses

We performed exactly the same analyses on the behavior of the virtual subjects and human participants. Trials in which the participant or virtual subject made no response were excluded, as were trials where the response or decision time was classified as an outlier by the median deviation of the medians applied method (*Rousseeuw and Croux, 1993*). Stimulation blocks with human participants were conducted in separate sessions, to avoid carry-over effects of repeated stimulation blocks, and compared with the directly preceding sham block in the same session (*Figure 6B*). We therefore have separate baselines for each stimulation condition, and consequently separate plots for depolarizing and hyperpolarizing conditions in *Figures 6* and *7*. In each analysis, we compared the sham blocks preceding the stimulation blocks in order to verify that they did not significantly differ from each other.

The accuracy of the virtual subject and human participant performance was measured as the percentage of trials in which the direction of coherent motion was indicated correctly, and each virtual subject and participant's accuracy threshold was defined as the motion coherence required to reach 80% accuracy. This was determined by fitting the percentage of correct trials at each coherence level to a Weibull function and taking the inverse of that function at 80%. Wilcoxon tests for comparing two repeated measures were used to compare the accuracy thresholds in each stimulation condition to the no stimulation condition in virtual subjects, and in each stimulation block to the preceding sham block in the same session in human participants. The decision time of the virtual subjects was analyzed by fitting the difference in mean decision times between the stimulation conditions and no stimulation condition at each coherence level to a linear function ($DT_{stim}$-$DT_{control}$= $\beta_0$+ $\beta_1 c$, where $DT$ is the decision time and $c$ is the coherence). The differences between the mean response times of human participants in stimulation blocks and the preceding sham block were also analyzed using linear regression. The two sham conditions (sham blocks directly preceding depolarizing blocks and those directly preceding hyperpolarizing blocks) did not differ from each other in accuracy threshold ($W(23)$ = 139, $p$=0.753) or response time difference ($B_1$ = −45.011, $p$=0.213).

Choice hysteresis was analyzed in two different ways. The first analysis involved splitting trials into two groups based on the decision made in the previous trial (Left*, and Right*), fitting the percentage of rightward choices in each group to a sigmoid function of the coherence to the left or right, and computing the difference in 'indecision points', or the level of coherence where rightward choices were made 50% of the time, between the two groups (*Padoa-Schioppa, 2013*; *Rustichini and Padoa-Schioppa, 2015*). In the second analysis, we modeled the decision as:

$$R = 1/(1 + e^{-X})$$
$$X = a_0 + a_1 c + a_2 (\delta_{n-1,R} - \delta_{n-1,L})$$

where $R$ is equal to one if right is chosen and 0 otherwise, and $c$ is the coherence level (negative if to the left, positive if to the right, therefore $a_1 > 0$) (*Padoa-Schioppa, 2013*; *Rustichini and Padoa-Schioppa, 2015*). The current trial is $n$, therefore $\delta_{n-1,R}$ is one if the choice in the last trial was rightward, otherwise 0, and $\delta_{n-1,L}$ is one if the choice on the last trial was leftward, otherwise 0. The term $\delta_{n-1,R}$- $\delta_{n-1,L}$ is thus −1 if the previous choice was leftward or one if it was rightward. Choice hysteresis is indicated by a value of $a_2$ greater than zero. We normalized the effect of hysteresis on the choice by the effect of coherence by analyzing the distribution of $a_2/a_1$ across virtual subjects for each condition. Wilcoxon tests for comparing two repeated measures were used to compare both the indecision point shift and coefficient ratio in each stimulation condition to the no stimulation condition in virtual subjects, and in each stimulation block to the preceding sham block in human participants. The two sham conditions (those directly preceding depolarizing blocks and those directly preceding hyperpolarizing blocks) did not differ from each other in terms of indecision point shift ($W(23)$ = 129, $p$=0.549) or logistic regression coefficient ratio ($W(23)$ = 123, $p$=0.441).

All data are archived on Dryad (*Bonaiuto et al., 2016*) and may be accessed via 10.5061/dryad.r1072. Python code to run analyses and generate figures from the manuscript is available on GitHub: https://github.com/jbonaiuto/perceptual-choice-hysteresis.

# Additional information

## Funding

| Funder | Grant reference number | Author |
|---|---|---|
| H2020 European Research Council | 260424 | James J Bonaiuto Sven Bestmann |
| Medical Research Council | | Archy de Berker |

The funders had no role in study design, data collection and interpretation, or the decision to submit the work for publication.

## Author contributions

JJB, Conception and design, Acquisition of data, Analysis and interpretation of data, Drafting or revising the article; AdB, SB, Conception and design, Analysis and interpretation of data, Drafting or revising the article

## Author ORCIDs
James J Bonaiuto, http://orcid.org/0000-0001-9165-4082
Archy de Berker, http://orcid.org/0000-0002-3460-7172

## Ethics

Human subjects: The study was performed in accordance with institutional guidelines for experiments with humans, adhered to the principles of the Declaration of Helsinki and was approved by the UCL Research Ethics Committee (reference number 5833/001). Participants gave their informed written consent before participating.

# Additional files

## Major datasets

The following dataset was generated:

| Author(s) | Year | Dataset title | Dataset URL | Database, license, and accessibility information |
|---|---|---|---|---|
| Bonaiuto JJ, de Berker A, Bestmann S | 2016 | Data from: Neural hysteresis in competitive attractor models predicts changes in choice bias with non-invasive brain stimulation | http://dx.doi.org/10.5061/dryad.r1072 | Available at Dryad Digital Repository under a CC0 Public Domain Dedication. |

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
