## [Decision Letter]

Thank you for submitting your article "Neural hysteresis in competitive attractor models predicts changes in choice bias with non-invasive brain stimulation" for consideration by *eLife*. Your article has been favorably evaluated by Sabine Kastner (Senior Editor) and three reviewers, one of whom, Richard Ivry (Reviewer #1), is a member of our Board of Reviewing Editors. The following individual involved in review of your submission has agreed to reveal their identity: Boris Burle (Reviewer #2).

Summary:

The reviewers find considerable merit in this study. This paper provides a nice blend of theoretical and experimental work to demonstrate the possible neural bases for one form of response bias, namely that observed from trial to trial. The authors employ a rather detailed biophysical model but the core idea here is quite simple-residual activity in the decision units will influence decisions on subsequent trials. This effect will be most pronounced on difficult decisions, mainly because easy decisions rise very quickly to threshold. The authors perform a tDCS experiment to show a similar pattern of behavior in humans, with a greater carryover effect following depolarized stimulation arrangement and reduced carryover following hyperpolarized arrangement.

All in all, this is a very solid work piece of work. The reviewers do think that the paper could push things a bit harder to probe the robustness of the model. There is little discussion about alternative models (other than variation in locus of modeled stimulation) and thus it is difficult to evaluate the goodness of their model-it is really evaluated at a qualitative level. In addition, there are some obvious opportunities here to look at predictions from the model and ask if these are supported behaviorally, either using published data or by conducting an additional experiment. We outline recommendations for revision below.

Essential revisions:

1) Simulation effect of inter-trial interval being extended. Obviously, the carryover effect should diminish with an increase in ITI. In fact, with their parameters, the effect of ITI might be quite large over a limited range (e.g., 500 ms to 5 s). We would like to see this simulated and tested in a new behavioral study (or use published data, if appropriate). No need to do tDCS here; a simple behavioral experiment will suffice. However, the current tDCS data could also be analyzed to examine this issue since you have a range of ITIs. For example, a median split into short ITI vs. long ITI. We recognize that since you kept the trial rate constant, ITI is confounded with RT. Moreover, the small ITI range will reduce sensitivity so this re-analysis is unlikely to provide a strong test.

2) The text suggests that the simulations were done with trial-by-trial stimulation whereas the tDCS conditions were blocked in the experiment. If this is correct, it is not obvious if the model would make the same predictions if the simulated stimulation was blocked. The authors should clarify this issue and, if not tested, run a blocked simulation to verify that the trial-by-trial effects persist under this condition. (Perhaps it is obvious from the model parameters that trial-to-trial effects do not persist over neighboring trials because of reset properties-if so, this should be made explicit.)

3) An alternative model could be tested by removing reciprocal inhibition between the decision units. There are accumulator models that do not make an assumption of this form (independent accumulators) and the finding of partial errors (e.g., the work of B. Burle and colleagues) suggest that there can be pronounced activation in both choices at time of response onset. Does removing the reciprocal inhibition change things in a significant way?

4) Text does not indicate if the participants made left/right decisions with two fingers of one hand (which one?) or with two fingers from different hands. While this may seem (and be) a trivial point, it also gets at a subtle issue/assumption with the model. The model assumes that both response options are affected by the stimulation. It is reasonable to assume that the level of interaction is the same for all possible response pairs? Would one expect unilateral stimulation to produce similar interactions if effectors were both contralateral to side of stimulation as when only one effector was contralateral to side of stimulation. The authors should clarify how responses were made and make explicit implications of this issue (even if to comment on why they think it isn't important or to comment on predictions to be derived from consideration of this issue).

---

## [Author Response]

*[…] All in all, this is a very solid work piece of work. The reviewers do think that the paper could push things a bit harder to probe the robustness of the model. There is little discussion about alternative models (other than variation in locus of modeled stimulation) and thus it is difficult to evaluate the goodness of their model-it is really evaluated at a qualitative level. In addition, there are some obvious opportunities here to look at predictions from the model and ask if these are supported behaviorally, either using published data or by conducting an additional experiment. We outline recommendations for revision below.*

In brief, we have:

- Conducted additional simulations and a control experiment addressing the effect of varying ITIs. The model suggested that for long ITIs, choice hysteresis effects should diminish, and this was indeed borne out in our experimental data;

- We have now implemented an alternative (accumulator) model, motivated by the suggestion made by one of the reviewers. While this model replicates the basic behavioral effects on RT and accuracy, it does not produce choice hysteresis.

*Essential revisions:*

*1) Simulation effect of inter-trial interval being extended. Obviously, the carryover effect should diminish with an increase in ITI. In fact, with their parameters, the effect of ITI might be quite large over a limited range (e.g., 500 ms to 5 s). We would like to see this simulated and tested in a new behavioral study (or use published data, if appropriate). No need to do tDCS here; a simple behavioral experiment will suffice. However, the current tDCS data could also be analyzed to examine this issue since you have a range of ITIs. For example, a median split into short ITI vs. long ITI. We recognize that since you kept the trial rate constant, ITI is confounded with RT. Moreover, the small ITI range will reduce sensitivity so this re-analysis is unlikely to provide a strong test.*

This is an excellent suggestion. We had not originally pursued this prediction of the model because, as the reviewers correctly point out, in our experiment the trial rate was constant and ITI was confounded with RT. Following this suggestion, we have since run additional model simulations (subsection “Assessing the impact of ISI on choice hysteresis”) and a follow-up experiment with a range of ITIs (subsection “Human Participants”).

When we ran the model with increasing interstimulus intervals (ISIs, we use this measure rather ITIs as the model activity depends specifically on ISIs), choice hysteresis effects indeed diminished with longer ISIs (subsection “Perturbation of an attractor network modulates choice hysteresis”, last paragraph). We then tested an additional group of human participants with the same task, but with short (1.5s) or long (5s) ISIs, mixed within blocks. In accordance with the model predictions, we found that choice hysteresis effects were largely absent in trials following a 5s delay (subsection “Stimulation over human dlPFC directionally influences choice hysteresis in the same way as stimulation of a competitive attractor network”, last paragraph), but were similar to our initial experiment for short ISIs. This result was predicted by the model, and lends further support for the link between behavioral choice hysteresis and neural hysteresis, as derived from our model.

*2) The text suggests that the simulations were done with trial-by-trial stimulation whereas the tDCS conditions were blocked in the experiment. If this is correct, it is not obvious if the model would make the same predictions if the simulated stimulation was blocked. The authors should clarify this issue and, if not tested, run a blocked simulation to verify that the trial-by-trial effects persist under this condition. (Perhaps it is obvious from the model parameters that trial-to-trial effects do not persist over neighboring trials because of reset properties-if so, this should be made explicit.)*

In our simulations, stimulation was also blocked in order to replicate human experimental conditions. We have made adjustments to the Introduction and Materials and methods sections to make this clear, with thanks.

*3) An alternative model could be tested by removing reciprocal inhibition between the decision units. There are accumulator models that do not make an assumption of this form (independent accumulators) and the finding of partial errors (e.g., the work of B. Burle and colleagues) suggest that there can be pronounced activation in both choices at time of response onset. Does removing the reciprocal inhibition change things in a significant way?*

This is a very interesting question, and initially we thought that accumulator models would make similar predictions. Because the pyramidal populations inhibit each other via the common pool of interneurons, it is not possible to directly remove mutual inhibition. We therefore constructed an accumulator version of our model by splitting the inhibitory interneuron populations into two subpopulations, each connected only with the corresponding pyramidal population (subsection “Control simulations: Accumulator with independent interneuron pools”, first paragraph). This way the pyramidal populations could not exert any inhibitory influence on each other. We found that the model made predictions similar to those made by the competitive attractor model in terms of the effects of stimulation on accuracy and decision time, but it did not exhibit any choice hysteresis behavior. This was because without mutual inhibition between the two pyramidal populations, activity from both pyramidal populations carried over to the next trial, rather than just activity from the winning population. This reduced the net bias exerted on the decision since both pyramidal populations were similarly biased. We sought to keep the accumulator version of the model as similar to the competitive attractor version as possible to ensure a fair comparison. However, the accumulator version was highly sensitive to the background firing rate and response threshold levels and we therefore had to restrict the ranges of these parameters when instantiating virtual subjects. It is therefore still possible that some other set of model parameters would give rise to choice hysteresis, but this would be beyond the scope of this paper. We also note that removing inhibition in a different way, by removing the projections from the pyramidal populations to the inhibitory population caused the network to become unstable and extremely sensitive to its inputs.

*4) Text does not indicate if the participants made left/right decisions with two fingers of one hand (which one?) or with two fingers from different hands. While this may seem (and be) a trivial point, it also gets at a subtle issue/assumption with the model. The model assumes that both response options are affected by the stimulation. It is reasonable to assume that the level of interaction is the same for all possible response pairs? Would one expect unilateral stimulation to produce similar interactions if effectors were both contralateral to side of stimulation as when only one effector was contralateral to side of stimulation. The authors should clarify how responses were made and make explicit implications of this issue (even if to comment on why they think it isn't important or to comment on predictions to be derived from consideration of this issue).*

We thank the reviewers for making this important point. Participants responded using only their right hand, which was thus contralateral to the site of stimulation, and stimuli were presented in the center of the visual field. This is the same as in Philiastides et al. (2011) who also targeted the left dlPFC with TMS during perceptual decision making. Our study therefore cannot address potential bilaterality of dlPFC organization. However, in nonhuman primates, dlPFC neurons are active during perceptual decision making whether the decision is indicated with a button press (Hussar & Pasternak, 2013) or a saccade (Kiani, Cueva, Reppas, & Newsome, 2014; Kim & Shadlen, 1999; Opris & Bruce, 2005). In humans, the left dlPFC is active during tasks using right-handed (Donner et al., 2007; Heekeren, Marrett, Bandettini, & Ungerleider, 2004; Heekeren, Marrett, Ruff, Bandettini, & Ungerleider, 2006; Philiastides & Sajda, 2007; Ruff, Marrett, Heekeren, Bandettini, & Ungerleider, 2010) or left-handed (Pleger et al., 2006) button presses, as well as saccades (Heekeren et al., 2006) to indicate the choice. Wenzlaff et al. (2011) used a perceptual decision making task in which subjects indicated one choice with the left hand and the other with the right and found that the left dlPFC reflected accumulated sensory evidence. This suggests that the left dlPFC is involved in perceptual decision making independent of response modality (e.g. it is activated when responding with the eyes and with the hands) or laterality (e.g. it is involved in decisions indicated with both the left and right hand). We are therefore confident that stimulation of left dlPFC would affect both hands equally, but we have added a discussion of this point to the subsection “Study Limitations” (fourth paragraph).